# Bitstream-Corrupted Video Recovery:
# A Novel Benchmark Dataset and Method

**Tianyi Liu**[1][*] **Kejun Wu**[1][*] **Yi Wang**[2][‡] **Wenyang Liu**[1] **Kim-Hui Yap**[1][†] **Lap-Pui Chau**[2]

[1]School of EEE, Nanyang Technological University, Singapore
[2]Department of EEE, The Hong Kong Polytechnic University, Hong Kong
{liut0038, wang1241, wenyang001}@e.ntu.edu.sg,
{kejun.wu, ekhyap}@ntu.edu.sg, lap-pui.chau@polyu.edu.hk

## Abstract

The past decade has witnessed great strides in video recovery by specialist technologies, like video inpainting, completion, and error concealment. However, they typically simulate the missing content by manual-designed error masks, thus failing to fill in the realistic video loss in video communication (e.g., telepresence, live streaming, and internet video) and multimedia forensics. To address this, we introduce the bitstream-corrupted video (BSCV) benchmark, the first benchmark dataset with more than 28,000 video clips, which can be used for bitstream-corrupted video recovery in the real world. The BSCV is a collection of 1) a proposed three-parameter corruption model for video bitstream, 2) a large-scale dataset containing rich error patterns, multiple corruption levels, and flexible dataset branches, and 3) a new video recovery framework that serves as a benchmark. We evaluate state-of-the-art video inpainting methods on the BSCV dataset, demonstrating existing approaches' limitations and our framework's advantages in solving the bitstream-corrupted video recovery problem. The benchmark and dataset are released at https://github.com/LIUTIGHE/BSCV-Dataset.

## 1 Introduction

As Cisco's report [1] shows, video traffic is expected to account for 82% of all internet traffic by 2022, making it the commonest multimedia type on the internet. However, due to unreliable channels and physical damage of the storage medium, videos are vulnerable to generated errors in the case of packet loss during transmission and context corruption during compression and storage [2]. Meanwhile, malicious attacks on the video decoder ecosystem may cause the risk of severe damage to video bitstreams [3]. Therefore, bitstream damage during compression, storage, and transmission chains is a common and crucial problem. The various types of damage factors yield different corruption degrees and error patterns in decoded frames, which are irreversible and unpredictable. Recovering the video content in corrupted bitstreams is of vital importance but beset with difficulties.

Researchers have been dedicated to video recovery at the encoding, transmission, and decoding stages [4]. Reed-Solomon codes [5] add redundant information during the encoding stage to enable error correction for the receiver. Checksum [6] is used in the transmission process to detect errors and

---

[*]Equal first contribution

[†]Corresponding author

[‡]Yi Wang was with NTU when this research was conducted and is currently with Hong Kong PolyU.

Submitted to the 37th Conference on Neural Information Processing Systems (NeurIPS 2023) Track on Datasets and Benchmarks. Do not distribute.

**Manual-Designed Corrupted Video**          **Real Bitstream-Corrupted Video**

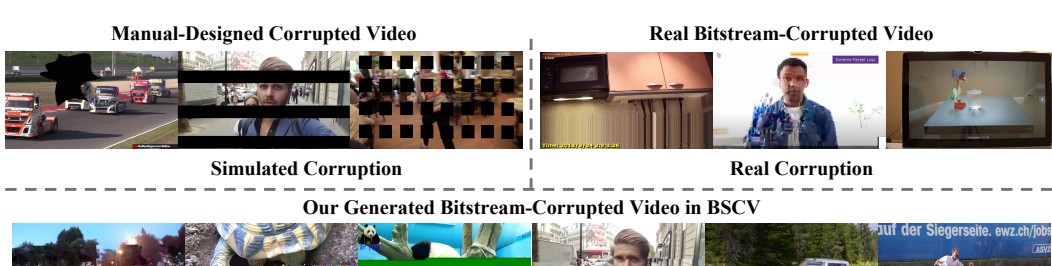

**Simulated Corruption**          **Real Corruption**

**Our Generated Bitstream-Corrupted Video in BSCV**

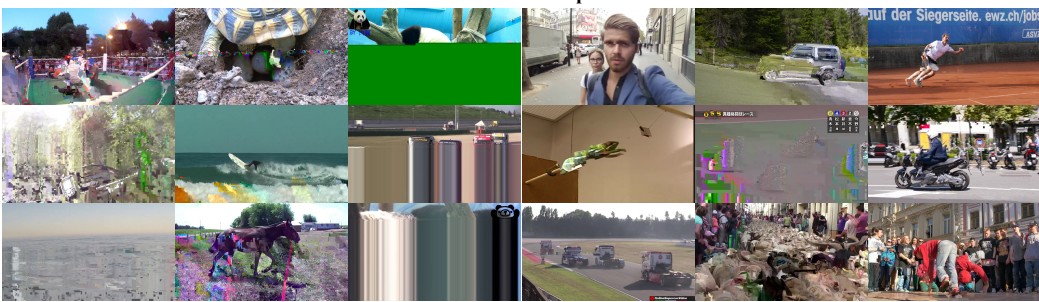

(1)Blocking Artfs.   (2)Color Artfs.   (3)Duplication Artfs.   (4)Misalignment   (5)Texture Loss   (6)Trailing Artfs.

Figure 1: Summary of the corruption pattern in video recovery problem. Compared with the simulated video corruption in existing inpainting or error concealment reseach, our dataset contains various realistic corruption patterns including (**1**) block artifacts (artfs.), (**2**) color artifacts, (**3**) duplication artifacts, (**4**) misalignment, (**5**) texture loss, (**6**) trailing artifacts, which is closer to the corrupted videos[1,2,3] in real world.

initiate re-transmission when errors are detected. These methods introduce additional requirements and inflexibility in hardware design and system reliability, and they cannot deal with long sequence loss in bitstreams. More research has focused on visual-based solutions in the decoding stage due to its intuitive and easy access to images, such as error concealment, completion, frame interpolation, and video inpainting.

Typically, the error concealment is to mitigate the effect of errors on video quality [7]. However, the error patterns are generally simulated by error masks of the slice or block shapes directly on the decoded video content. The fixed and simple error simulation limits the application scenarios, as the error patterns in realistic scenarios are neither fixed nor simple. Frame interpolation [8, 9, 10, 11, 12] is another visual-based solution. It synthesizes intermediate frames from a given set of correctly decoded frames to replace damaged or lost frames. Nonetheless, interpolation methods are barely satisfactory when there exists a large scene motion between frames [13], and when they encounter errors spread across a sequence of frames. Video inpainting is similar to video completion, which aims to complete missing regions in a given video [14]. Generally, video inpainting takes the surrounding temporal and spatial content as a reference to fill corrupted regions by learning underlying patterns and structural features of videos [15, 16]. However, the corrupted regions are commonly simulated by user-predefined binary error masks instead of natural errors generated from the real bitstream.

For the application scenarios of video storage, communication, and internet video, the manually created masks have difficulty in reflecting the shapes and patterns of real corruption. The requirements of video content coherence in temporal and spatial dimensions are hard to meet when large motion and details are missing across frames [15]. Therefore, the real bitstream and video datasets, as well as video content recovery methods, are highly necessary and urgently needed. So far, there is no large-scale dataset specialized for bitstream-corrupted video recovery. Existing inpainting datasets are limited to simulated error masks, and error concealment datasets are small-scale and may require extracting motion information from bitstream, which is not always available [17].

---

[1]https://blog.quindorian.org/2013/03/fixing-rtsp-stream-corruption.html/
[2]https://www.youtube.com/watch?app=desktop&v=M7wZxk7yPeQ
[3]https://www.youtube.com/watch?v=l66kIS_-UmI

In this paper, we construct the first large-scale benchmark to facilitate the research of bitstream-corrupted video (BSCV) recovery. Our BSCV dataset includes more than 28,000 bitstream-corrupted video clips (over 3,500,000 frames), which are extracted and elaborated from the most popular video inpainting datasets, YouTube-VOS [18] and DAVIS [19]. Specifically, we compress these video clips into bitstreams using the most popular H.264 video codec [20]. Segments in bitstreams are randomly removed to simulate the effect of packet loss error and storage damage error on decoded videos, and these error types are common in real-world multimedia communications [21]. The simulated error patterns used in typical video recovery tasks and the real error patterns of our dataset are shown in Fig. 1. It can be observed that the video error types in our dataset are sophisticated and unpredictable, while others are simple and fixed. Therefore, our dataset enables us to reveal the problem of real-world video corruption in multimedia communications completely. Furthermore, we also provide a specialized recovery method for bitstream-corrupted video. The remaining semantic information in corrupted regions is incorporated with the spatially and temporally adjacent information to recover the corrupted regions.

The main contributions are as follows: (i) We construct BSCV, the first large-scale dataset used for bitstream-corrupted video recovery in the real world. The provided videos are decoded from real corrupted bitstreams, which are generated by our three-parameter bitstream corruption model. The dataset contains over 28,000 challenging corrupted video clips with realistic and unpredictable error patterns, multiple corruption levels, and flexible dataset branches. (ii) We propose a new video recovery framework inspired by video inpainting pipeline. It completes and enhances the feature representation capability by extracting residual visual information from the corrupted region, achieving higher recovery quality. (iii) We perform a comprehensive evaluation of our dataset to reveal the limitation of existing video inpainting algorithms and point out the future direction.

## 2 Related Works

**Benchmark and dataset.** To the best of our knowledge, currently, there is no bitstream-corrupted video benchmark for the research of bitstream-corrupted video recovery.

As shown in Table 1, for conventional error concealment and corruption recovery research, using a small set of YUV sequence [22] to test the algorithm performance is a common practice. In that case, researchers usually simulate different kinds of stripe or packet loss-caused masks on those video sequences [25, 26, 27, 7]. However, the scale issue limits the application of deep learning methods. Along with the development of video datasets for different computer vision applications, some datasets such as

| Dataset Name | Clip Numbers | Bitstream Provided | Bitstream Corruption | Mask Provided | Application Scenarios |
|---|---|---|---|---|---|
| YUV Sequence [22] | 26 | × | × | × | VC, VR |
| Vimeo90K [23] | 90,000+[1] | × | × | × | SR, ITP |
| REDS [24] | 300 | × | × | × | SR, DB |
| DAVIS [19] | 150 | × | × | ✓ | OS, INP |
| YouTube-VOS [18] | 4,000+ | × | × | ✓ | OS, INP |
| BSCV(Ours) | 28,000+ | ✓ | ✓ | ✓ | VR, SR, ITP, INP |

[*] VC: Video Coding, SR: Super Resolution, ITP: Interpolation, DB: Deblurring, OS: Object Segmentation, INP: Inpainting, VR: Video Recovery
[1] Frame numbers are fixed at 7 frames.

Table 1: Comparisons among video datasets

Vimeo90K [23], REDS [24] are proposed for video restoration tasks including super-resolution, deblurring, and so on. Recently, deep learning-based video completion assumes a very similar task setting with error concealment which is a sub-task of video inpainting. By accepting arbitrary masks, learning-based video inpainting can be trained by a large number of samples. The setting of the mask is usually a fixed mask [14] or an object-like mask with limited size, random shape, and motion [16, 28, 15, 29]. Most datasets involve the content of the videos in the DAVIS [19] and YouTube-VOS [18]. However, DAVIS is still a relatively small scale with only 150 video clips, and therefore it is usually used for qualitative evaluation in video inpainting research. Recently, YouTube-VOS has been a widely-used large-scale dataset for various video inpainting algorithm training because of its content diversity. Nevertheless, along with recent research of bitstream-corrupted image restoration [30], the large-scale video dataset never considers such video corruption. With the simulated mask setting, video inpainting and different kinds of video recovery research are difficult to perform well in complex and unpredictable video corruptions because of the gap between the human-predefined binary mask and unpredictable mask supervision. Besides, modern video datasets

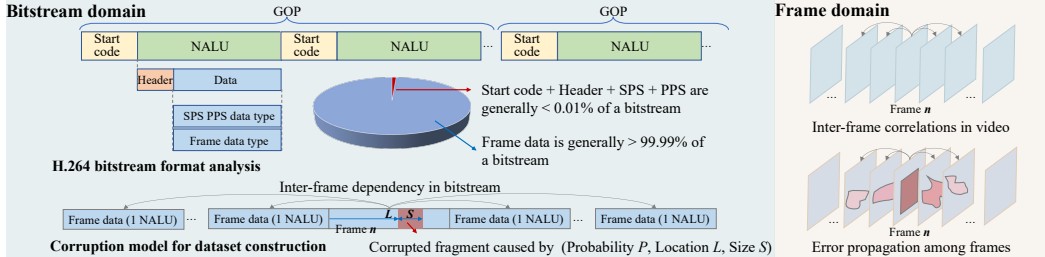

Figure 2: Left: H.264 bitstream statistics and the proposed corruption model. Right: Inter-frame correlations and error propagation in the frame domain.

are usually packed in frame sequences, bitstream-related research still hungers for data in the current deep learning era.

**Video restoration.** Restoring image and video data in various visual environments [31, 32, 33] is currently a key application of artificial intelligence. As videos can be treated as multiple consecutive images/frames, earlier works [34, 35] simply reuse the ideas from image restoration with the temporal redundancy of neighboring frames fails to be explored. To fully utilize temporal information, Xue *et al.* [23] proposed a task-oriented flow to achieve feature alignment explicitly. Other studies utilized dynamic upsampling filter [36] or deformable convolution [37] to achieve implicit motion compensation. As for feature fusion, either a one-stage direct fusion structure [37, 38] or multi-stage progressive fusion structure [39] have been used in existing methods.

**Video error concealment.** Video error concealment, a commonly-used post-processing technique at the decoder side, aims to recover the error regions in decoded videos [26]. It can be divided into various categories in the bitstream and pixel levels, including spatial, temporal, and hybrid spatial-temporal methods [17, 7, 40]. Traditionally, at the bitstream level, missing motion information can be estimated by surrounding motion vector and block partitioning in the previous frame [41, 42]. At the pixel level, pixel-wise processing is capable but relies on deficient spatial information [17]. Recently, deep learning-based methods still assume a traditional corruption pattern and use experimental mask settings to simulate stripe or patch loss [43, 44, 26, 25]. It makes these methods not suitable for recovering bitstream-corrupted videos because the corruption caused by realistic packet loss is generally unpredictable and irregular.

**Video inpainting/completion.** Video inpainting is to generate content in unfilled regions of a video, which accepts arbitrarily defined masks to indicate corrupted regions. Traditionally, video inpainting is considered as a patch matching or pixel diffusion problem [45, 45, 46, 47, 48]. In the era of deep learning, the patch-based method also makes significant success [16, 49]. Flow-guided generative methods are currently mainstream in video inpainting, leveraging motion information for spatial and temporal relationships between frames [14, 50, 28, 51, 52, 15, 53]. DFVI [14] was a pioneering work that formulates the generative video inpainting problem as a pixel propagation task rather than simply filling RGB values in corrupted regions. Li *et al.* [15] built the traditional 3-stage video inpainting pipeline optimized jointly and achieved an efficient end-to-end framework for video inpainting. In the context of bitstream-corrupted video recovery, video inpainting is closely related. However, existing research often overlooks the performance of inpainting algorithms when dealing with dynamic and large masks. Consequently, they fail to address complex recovery scenarios with significant corrupted areas and partially remained content caused by bitstream corruption.

## 3 Bitstream-corrupted Video Dataset

**H.264 bitstream and bitstream corruption.** The most popular video codec, H.264, was used by 85% of video developers in 2022 [54]. The compatibility of H.264 with a variety of devices and platforms empowers the delivery of video content bitstream over the internet. The H.264 bitstream

domain is shown in Fig. 2. The typical format of H.264 bitstream consists of successive NALUs (network abstraction layer units). A bitstream contains several bytes of start code prefix and Header. The SPS (sequence parameter sets) and PPS (picture parameter sets) also occupy a small number of bytes. By investigating the bitstream component, we find that the bytes of SPS, PPS, header, and start code only take up a negligible proportion, e.g., 0.01% on example bitstream. In contrast, the frame data occupies a dominant proportion of a H.264 bitstream.

The bitstream segments and packets are possibly corrupted or lost in the chains of video storage, encoding, transmission, and decoding. Therefore, video recovery from corrupted bitstreams is in surging demand. Due to the significant proportion of frame data in a bitstream file, corruption is most likely to occur in the frame data parts, which is the basic assumption in this paper. A frame can generally refer to other previously coded frames for high coding efficiency [55, 56]. As shown in the frame domain of Fig. 2, there exist inter-frame correlations among frames of a video when encoding a video into the bitstream. The inter-NALU dependencies are accordingly created in a bitstream. Thus, error propagation among frames tends to be irregular and unpredictable.

**Bitstream-corrupted video generation.** Due to the popularity of H.264, video clips are encoded by H.264 codec to generate bitstream files of these videos. The coding configuration selects widely used close-GOP (group of pictures), and the GOP sizes adopt 16 frames for a long-range reference. We simulate the corruption pattern by removing specific segments of some NALUs in a bitstream, as shown in the bottom part of Fig. 2. The extremely low proportions of the start code, NALU headers, SPS, and PPS yield a extremely low corruption probability on these parts. Furthermore, corruption on these parts may cause severe errors, e.g., even decoding failures, which is out of vision research. Therefore, we mainly focus on the remaining bitstream of frame data with dominant proportion. Based on the analysis, we can randomly corrupt frames in visual level. Therefore, we parameterize a three-parameter corruption model $(P, L, S)$, where the corrupted fragments are defined by frame corruption probability $P$, corruption location $L$, and fragment size $S$.

The corrupted bitstreams are parsed and decoded by H.264 decoder, generating videos with unpredictable regional errors.

**Dataset construction and statistics.** By the above bitstream corruption, we construct a bitstream-corrupted video (BSCV) dataset based on two commonly used datasets in video inpainting, i.e., YouTube-VOS [18] and DAVIS [19]. In detail, we extract 4,132 original videos from YouTube-VOS and DAVIS datasets to generate the main branch.

DAVIS provides 480P videos with dense object segmentation annotation which is mostly used for evaluation in prior works, we followed its application in this paper as well. YouTube-VOS is mainly a 720P dataset which also contains several 1080P videos. Our method is also applicable for those 1080P videos which proves its scalability. Then we additionally provide an additional 1080P branch with 256 videos from YouTube-UGC dataset [57]. It contains longer frame sequence and higher resolution which could enrich the source of our dataset, allowing further extension based on it. Further, we provide a small 4K branch using videos from Videezy4K [58] as an reference example for the future extension to higher resolution videos.

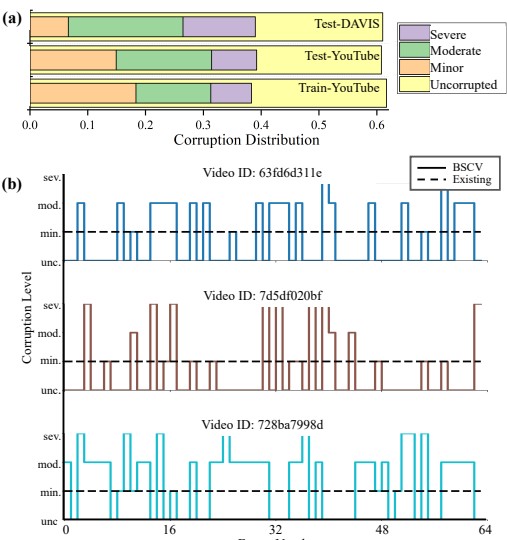

Figure 3: Taking the BSCV dataset branch in the parameters of $(1/16, 0.4, 4096)$ as an example for illustration. (a) The statistics of corruption distribution. (b) The corruption level changes among frames for some sampled videos.

By setting the parameter combinations of the proposed corruption model, multiple branches of the BSCV dataset can be generated. We also provide error region masks in the dataset. Specifically, grayscale difference maps are calculated by subtracting the corrupted videos from the corresponding original videos decoded from the corruption-free bitstream. The slight changes below the default threshold are suppressed, and the small outliers inside or outside masks are removed by morphological filtering.

For the BSCV dataset branch in the parameters of $(1/16, 0.4, 4096)$, Fig. 3 illustrates the corruption statistics of the branch and the corruption degree of randomly selected video samples. The area ratio of corrupted regions to their corresponding frame is referred to as the "corrupted area ratio". The ratios range from 0-10%, 10-30%, and above 30% are defined as minor (min.), moderate (mod.), and severe (sev.) corruption levels. The ratio of 0 is defined as corruption-free (unc.). We observe that nearly 30% of frames are corrupted for this example dataset branch. Compared with the existing video inpainting tasks with fixed mask area settings (e.g., 1/16), the frame corruption in our dataset is complex, variable, and unpredictable, making it closer to realistic scenarios and more challenging.

We further analyze the rich error patterns of our dataset shown in Fig. 1. Color artifacts occur when chrominance information is corrupted, which is more severer than edge color bleeding in typical compression artifacts [59]. The trailing artifacts come from the corruption of motion information, which causes a floating trailing effect in subsequent frames. The texture information corruption and error propagation may cause blocking artifacts. Duplicate artifacts are common in intra-coding regions, which duplicate the error pixels in the adjacent regions. More details on dataset construction and analysis of error patterns can refer to the Supplementary Material.

**Flexibility and extensibility.** The constructed dataset and proposed three-parameter corruption model can provide flexibility in dataset customization and extensibility in application scenarios. By setting different parameter combinations, it is flexible to construct custom datasets to meet specific application scenarios, which is demonstrated in the experiment section. We also developed a video recovery framework without relying on the motion, partition, and residual information in case they are not available in a corrupted bitstream. Thus, the provided dataset and recovery framework can extend to broad bitstream corruption scenarios, such as packet loss during transmission, segment corruption in compression, and deletion of partial data in storage. The application scenarios are not limited to bitstream-related video recovery. It is also suitable for local and cloud video processing tasks, like video inpainting, completion, and manipulation.

## 4 Bitstream-corrupted Video Recovery Framework

In this section, inspired by video inpainting framework, we propose a specialized video recovery method achieves feature completion through segmented feature extraction and fusion, thereby better coordinating with optical flow information to guide the generation of high-quality recovery content.

The overview of the proposed bitstream-corrupted video recovery (BSCVR) framework is illustrated in Fig. 4. We propose to enable an additional perception channel to video inpainting frameworks. It extracts and fuses local features from corrupted and corruption-free regions. By encoding the residual information inside the corrupted regions into the local features, it can greatly enhance the feature completion and representation capability compared to existing video inpainting frameworks. Consequently, the enhanced feature can provide a solid reference for the subsequent recovery process. Then a flow-guided feature propagation module is used in [15] to propagate the content. Combining with reference content from non-local frames, the content generation module is implemented by stacking several temporal focal transformers [15].

To be specific, given a corrupted video frame sequence $\{X^t \in \mathcal{R}^{3 \times h \times w} | t = 1, 2, ..., T\}$, and its corresponding mask sequence indicating the corrupted regions $\{M^t \in \mathcal{R}^{1 \times h \times w} | t = 1, 2, ..., T\}$. The video recovery framework is expected to recover the corrupted region with spatially and temporally plausible content. According to Fig. 4, for the input corrupted frame sequence, we use a context encoder $(E)$ [60] to perform region-based encoding. $\{Q^t \in \mathcal{R}^{3 \times h \times w} | t = 1, 2, ..., T_l\}$ indicated

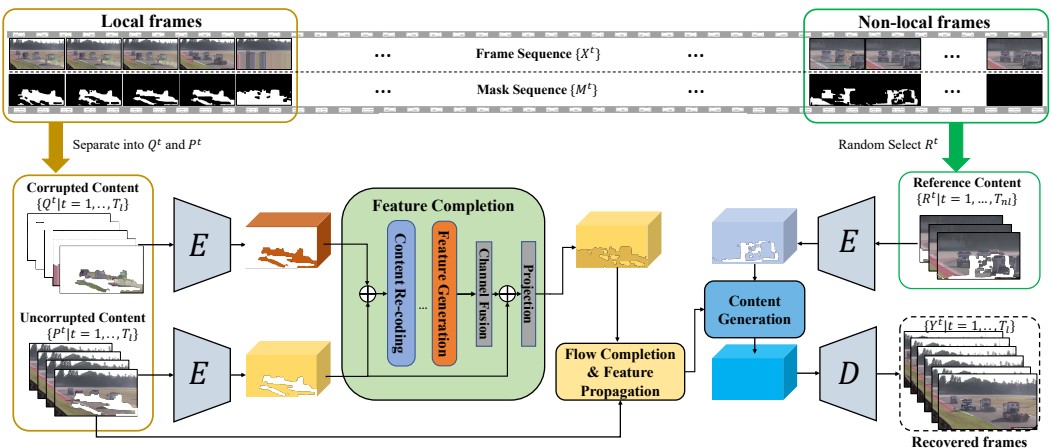

Figure 4: Overview of our bitstream-corrupted video recovery (BSCVR) framework. Compared with existing methods, we follow the common practice by inputting the corruption-free content as the basic information source when constructing local features for recovery. We additionally enable a new input channel for the corrupted region and extract the feature of its partial contents which is completely ignored by existing methods. With transformer-based architecture, the local feature can be completed and enhanced by encoding the feature of partial contents into it.

by masks will be separately input into the recovery framework. Then, we propose to use several transformer encoder layers [61] to fuse and re-encode these two features to achieve feature completion. By attention-based decoding and channel fusion, an intermediate feature is generated. Consequently, with skip connection and output projection, the representative capability of the resulting feature can be further enhanced by fusing multi-scale and multi-level information. We then follow the approach of flow-guided video inpainting to extract and complete optical flows from neighboring frames to serve as guidance for feature alignment and propagation. Afterward, a content generation module based on temporal focal transformer and soft splitting will combine the enhanced, aligned, and propagated features of local neighboring frames with the reference features of non-local frames' corruption-free regions $\left\{ R^t \in \mathcal{R}^{3 \times h \times w} | t = 1, 2, ..., T_{nl} \right\}$ to generate content and finally reconstruct a result frame sequence $\left\{ \hat{Y}^t \in \mathcal{R}^{3 \times h \times w} | t = 1, 2, ..., T_l \right\}$ through a decoder ($D$) module. More detailed descriptions of the methodology can be found in the Supplementary Material.

## 5 Experiment

The proposed BSCVR and state-of-the-art (SOTA) video inpainting methods are performed on the constructed BSCV dataset. We conduct comprehensive quantitative and qualitative evaluations to demonstrate the effectiveness of our dataset and method. The flexibility of our corruption model and the robustness of our BSCVR framework are validated on multiple branches of BSCV.

**Experimental setting.** We adopt the corruption parameter of $(1/16, 0.4, 4096)$, and its corresponding statistics have been illustrated in Fig. 3. This setting has moderate difficulties with adequate corruption types, which is suitable for video inpainting methods. The corrupted region is usually recoverable yet challenging. SOTA video inpainting methods are compared with our method to recover corruption-free videos. STTN [28] and FuseFormer [16] downsample videos to 240P due to the limitation of computational complexity. E2FGVI-HQ is an upgraded version of E2FGVI [15], stating that it could take arbitrary resolution and generate results with the original input resolution. They could be viewed as the main competitor of our method. Noted that previous works set the mask with "random shape and location" to augment inpainting data, and those pre-trained models are trained with 500K iterations. In contrast, training those methods on our dataset requires only 250K iterations to converge. The detailed implementation of our method can refer to the Supplementary Material.

| Test res. | Method | Accuracy | | | | | | | | Efficiency |
|---|---|---|---|---|---|---|---|---|---|---|
| | | YouTube-VOS (720P) Subset | | | | DAVIS (480P) Subset | | | | Runtime |
| | | PSNR↑ | SSIM↑ | LPIPS↓ | VFID↓ | PSNR↑ | SSIM↑ | LPIPS↓ | VFID↓ | s/frame |
| 240P | Input | 18.8749 | 0.8160 | 0.1527 | 0.2015 | 18.4562 | 0.7921 | 0.1541 | 0.4189 | - |
| | STTN [28] | 29.3840 | 0.9174 | 0.0465 | 0.0566 | 26.2172 | 0.8600 | 0.0638 | 0.1589 | 0.120 |
| | STTN [28]* | 29.9172 | 0.9303 | 0.0394 | 0.0544 | 26.5453 | 0.8722 | 0.0575 | 0.1534 | 0.120 |
| | FuseFormer [16] | 28.8012 | 0.9047 | 0.0549 | 0.0641 | 26.2547 | 0.8618 | 0.0659 | 0.1645 | 0.200 |
| | FuseFormer [16]* | 29.8108 | 0.9328 | 0.0381 | 0.0526 | 26.7367 | 0.8834 | 0.0531 | 0.1477 | 0.200 |
| | E2FGVI-HQ [15] | 29.6866 | 0.9228 | 0.0469 | 0.0555 | 26.7850 | 0.8765 | 0.0600 | 0.1513 | 0.160 |
| | E2FGVI-HQ [15]* | 31.0030 | 0.9473 | 0.0341 | 0.0479 | 27.6551 | 0.9018 | 0.0491 | 0.1387 | 0.160 |
| | **BSCVR-S (Ours)*** | 31.8345 | 0.9584 | 0.0262 | 0.0427 | 28.4211 | 0.9180 | 0.0381 | 0.1196 | 0.172 |
| | **BSCVR-P (Ours)*** | **31.9534** | **0.9598** | **0.0258** | **0.0426** | **28.5430** | **0.9199** | **0.0375** | **0.1165** | 0.178 |
| Ori-ginal | Input | 19.1490 | 0.8244 | 0.1415 | 0.0575 | 18.4384 | 0.7979 | 0.1490 | 0.1999 | - |
| | E2FGVI-HQ [15] | 28.5039 | 0.8783 | 0.0453 | 0.0126 | 25.7803 | 0.8236 | 0.0504 | 0.0468 | 0.192 / 0.176 |
| | E2FGVI-HQ [15]* | 29.5666 | 0.9023 | 0.3955 | 0.0161 | 26.6723 | 0.8611 | 0.0530 | 0.0577 | 0.192 / 0.176 |
| | **BSCVR-S (Ours)*** | **30.2235** | **0.9185** | 0.0335 | 0.0143 | **27.2770** | **0.8809** | 0.0427 | 0.0500 | 0.250 / 0.203 |
| | **BSCVR-P (Ours)*** | 29.9943 | 0.9144 | 0.0343 | **0.0104** | 26.3564 | 0.8511 | **0.0416** | **0.0406** | 0.261/ 0.213 |

Table 2: Quantitative results of SOTA pre-trained video inpainting methods, their corresponding models trained on our dataset (denoted by *), and our method. In our method, BSCVR-S means that the feature completion module considers the input feature as a sequence like traditional Transformer [61], and BSCVR-P indicates that the module considers the input as patches referring to SwinIR [31]. The comparison is conducted under the 240P setting due to the model capability of previous works. For the methods which are able to handle arbitrary-resolution video, we calculate metrics based on the original frame sequence, and we measured and demonstrate the runtime of the model under 720P (former) / 480P (latter) input, respectively.

**Evaluation metrics.** Regarding the quantitative evaluation, we measure the performance of inpainting algorithms based on two aspects: inpainting quality reconstruction and realism. Among them, Peak Signal-to-Noise Ratio (PSNR), Structural Similarity Index (SSIM) [62], and Learned Perceptual Image Patch Similarity (LPIPS) [63] using pre-trained AlexNet backbone [64] are mainly used to measure reconstruction performance. Video Fréchet Inception Distance (VFID) [65] is mainly used to measure performance in terms of realism. It corresponds to the sets of all recovered videos and all reference videos. The features are extracted from a pre-trained I3D backbone [66].

**Quantitative evaluation.** The quantitative results on the YouTube-VOS and DAVIS subsets are shown in Table 2. Due to the model capability of some previous works, we first follow the existing experimental setting of video inpainting to downscale the original video in our dataset to 240P and conducted model training and metric calculation. It can be observed that our methods achieve better results in all metrics. However, compromising on video resolution is not reasonable for the video recovery problem. Thus, we particularly compared our method with the SOTA method E2FGVI-HQ [15] which is currently the only method can handle

| Param. (P, L, S) | Methods | Accuracy | | | |
|---|---|---|---|---|---|
| | | DAVIS Subset | | | |
| | | PSNR↑ | SSIM↑ | LPIPS↓ | VFID↓ |
| (1/16, 0.4, 4096) | Input | 18.4384 | 0.7979 | 0.1490 | 0.1999 |
| | E2FGVI-HQ [15]* | 26.3734 | 0.8415 | 0.0466 | 0.0444 |
| | **BSCVR-S (Ours)** | **27.2770** | **0.8809** | 0.0427 | 0.0500 |
| | **BSCVR-P (Ours)** | 26.3564 | 0.8511 | **0.0416** | **0.0406** |
| (1/16, 0.4, **2048**) | Input | 18.2283 | 0.7798 | 0.1536 | 0.1920 |
| | E2FGVI-HQ [15]* | 26.0251 | 0.8423 | 0.4777 | 0.0440 |
| | **BSCVR-S (Ours)** | **26.2437** | **0.8554** | 0.0416 | 0.0401 |
| | **BSCVR-P (Ours)** | 26.1057 | 0.8525 | 0.0422 | 0.0407 |
| (1/16, **0.2**, 4096) | Input | 18.0789 | 0.7649 | 0.1569 | 0.2067 |
| | E2FGVI-HQ [15]* | 24.7468 | 0.7746 | 0.0656 | 0.0546 |
| | **BSCVR-S (Ours)** | **24.9832** | **0.7980** | 0.0570 | 0.0478 |
| | **BSCVR-P (Ours)** | 24.8303 | 0.7940 | 0.0579 | 0.0484 |
| (**2/16**, 0.4, 4096) | Input | 17.9418 | 0.7616 | 0.1592 | 0.1963 |
| | E2FGVI-HQ [15]* | 24.3774 | 0.7934 | 0.0623 | 0.0767 |
| | **BSCVR-S (Ours)** | **24.5066** | **0.8077** | 0.0553 | 0.0698 |
| | **BSCVR-P (Ours)** | 24.3808 | 0.8037 | 0.0561 | 0.0705 |
| (1/16, 0.4,**8192**) | Input | 18.6665 | 0.8170 | 0.1450 | 0.1849 |
| | E2FGVI-HQ [15]* | 26.0722 | 0.8371 | 0.0486 | 0.0455 |
| | **BSCVR-S (Ours)** | **26.2708** | **0.8518** | 0.0423 | 0.0417 |
| | **BSCVR-P (Ours)** | 26.1231 | 0.8487 | 0.0430 | 0.0418 |
| (1/16, **0.8**, 4096) | Input | 19.0062 | 0.8419 | 0.1389 | 0.1874 |
| | E2FGVI-HQ [15]* | **32.8311** | **0.9506** | 0.0162 | 0.0264 |
| | **BSCVR-S (Ours)** | 32.7959 | 0.9514 | **0.0147** | **0.0247** |
| | **BSCVR-P (Ours)** | 32.7204 | 0.9509 | 0.0149 | 0.0252 |
| (**4/16**, 0.4, 4096) | Input | 17.8542 | 0.7587 | 0.1610 | 0.1973 |
| | E2FGVI-HQ [15]* | **22.7912** | 0.7480 | 0.0738 | 0.1192 |
| | **BSCVR-S (Ours)** | 22.7094 | **0.7570** | **0.0679** | **0.1079** |
| | **BSCVR-P (Ours)** | 22.5480 | 0.7527 | 0.0686 | 0.1079 |

Table 3: Performance comparison with E2FGVI-HQ on different branches with varied corruption parameter combinations.

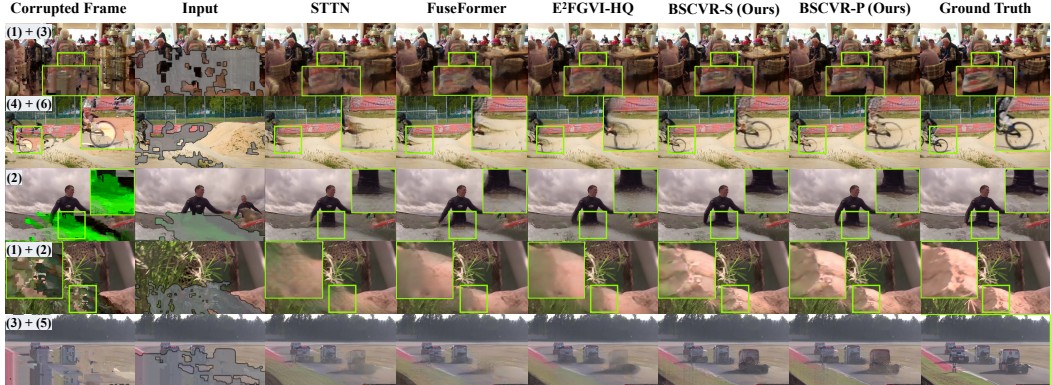

(a) Qualitative results comparing our method with STTN [28], FuseFormer [16], and E2FGVI-HQ [15] under unified 240P resolution.

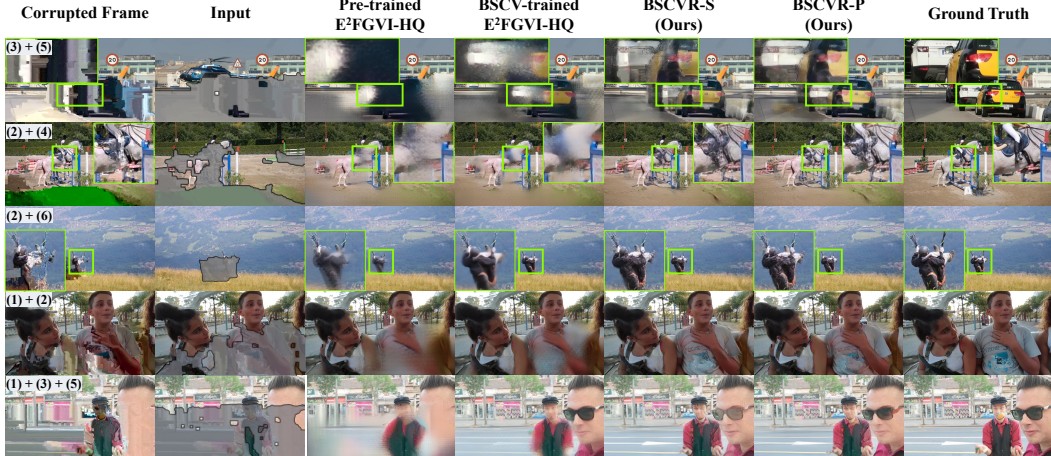

(b) Qualitative results comparing our method with E2FGVI-HQ [15] trained on our dataset under original video resolution.

Figure 5: Qualitative comparison of our method and SOTA video inpainting methods on low (a) and high (b) resolutions. The involved corruption types include **(1)** blocking artifacts, **(2)** color artifacts, **(3)** duplication artifacts, **(4)** misalignment, **(5)** texture loss, **(6)** trailing Artifacts and their combinations.

the original resolution scenario. For the metrics of PSNR, SSIM, LPIPS, and VFID, our method can achieve significant improvement, which makes the bitstream-corrupted video recoverable, e.g., >30dB PSNR for YouTube-VOS. Our method could comprehensively refine the content in corrupted regions to guide plausible content generation and keep low computational complexity by applying efficient model architecture referring to E2FGVI-HQ [15]. For the results on different corruption parameters and more comparison with latest non-end-to-end methods, we provided more experiments results and discussion in the Supplementary Material.

**Qualitative Evaluation.** For qualitative evaluation, we choose STTN [28], FuseFormer [16], and E2FGVI-HQ [15] trained on our dataset as comparison methods. The evaluation is conducted under 240P and original resolutions. Some representative corruption patterns and their recovery results are visualized in Fig. 5a. The comparison methods are difficult to generate plausible content to recover the corrupted region, while our proposed method can generate clearer details with more informative textures and structures. Our method and dataset limits the tendency of object removal when the mask is relatively large, especially under the original resolution and the compared results of our method with the SOTA method E2FGVI-HQ [15] are shown in Fig. 5b. These results demonstrate

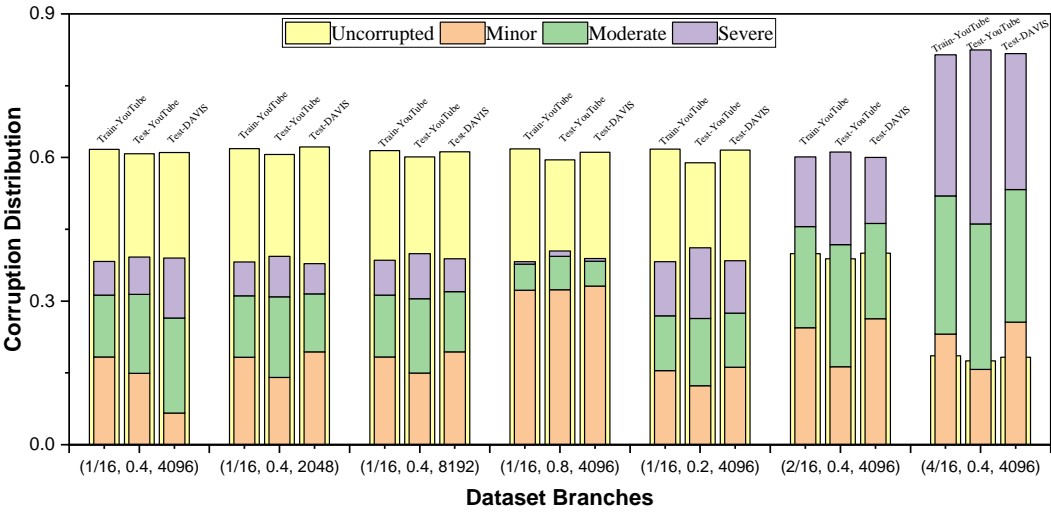

Figure 6: Corruption ratio distribution of the YouTube-VOS&DAVIS dataset branches for experiments under different corruption parameter combinations.

the limitation of current methods on the proposed dataset and the advantage of our high quality data and method. More visualized results can be found in the Supplementary Material.

**Flexibility in Dataset Construction.** We use the three-parameter corruption model $(P, L, S)$ in our dataset construction. We validate the model by generating more branches of dataset with seven additional parameter settings: $(1/16, 0.4, 2048)$, $(1/16, 0.4, 8192)$, $(1/16, 0.4, 4096)$, $(1/16, 0.2, 4096)$, $(1/16, 0.8, 4096)$, $(2/16, 0.4, 4096)$ and $(4/16, 0.4, 4096)$. These settings represent varying corruption probabilities, where $P = m/l$ implies that the corruption happens in random $m$ frames out of $l$ frames of a GOP.

We analyze the corruption distribution by calculating the ratio of the corrupted region to the frame resolution, as shown in Fig. 6. It indicates that parameter $P$ has the most significant impact on corruption, leading to a higher number of corrupted frames and more severe damage. For these dataset branches, we compare our BSCVR and E2FGVI-HQ under the original resolution on the DAVIS 480P subset. The results are listed in Tab. 3. It shows that the proposed BSCVR consistently outperforms E2FGVI-HQ, validating the robustness of our BSCVR on multiple dataset branches.

# 6   Conclusion

Aiming at the challenging problem of bitstream-corrupted video recovery in the real world, we construct the first large-scale benchmark, BSCV. The BSCV provides a bitstream corruption model, a realistic decoded video dataset, and a video recovery framework, BSCVR. The bitstream corruption model enables to flexibly generate dataset branches by specifying parameter combinations. The dataset contains 28,000 realistic video clips decoded from corrupted bitstreams with unpredictable error patterns and corruption levels. The BSCVR offers a effective framework for high-quality video recovery. Extensive experiments demonstrate that the proposed BSCVR outperforms SOTA video inpainting methods quantitatively and qualitatively. The flexibility of dataset construction and the robustness of our BSCVR framework are also validated on various dataset branches. The benchmark dataset is expected to benefit video recovery in multimedia forensics and video communication application includes live streaming and online conference, etc. The future work will concentrate on designing more reasonable bitstream corruption models, engaging more dataset sources, and creating more effective recovery frameworks.

## Acknowledgement

This research is supported by the National Research Foundation, Singapore, and Cyber Security Agency of Singapore under its National Cybersecurity Research & Development Programme (Cyber-Hardware Forensic & Assurance Evaluation R&D Programme <NRF2018NCRNCR009-0001>). Any opinions, findings and conclusions or recommendations expressed in this material are those of the author(s) and do not reflect the view of National Research Foundation, Singapore and Cyber Security Agency of Singapore.

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
