# Bitstream-Corrupted Video Recovery:
# A Novel Benchmark Dataset and Method
## *- Supplementary Materials -*

**Tianyi Liu**[1*]  **Kejun Wu**[1*]  **Yi Wang**[1,2‡]  **Wenyang Liu**[1]  **Kim-Hui Yap**[1†]  **Lap-Pui Chau**[2]

[1]School of EEE, Nanyang Technological University, Singapore
[2]Department of EiE, The Hong Kong Polytechnic University, Hong Kong
{liut0038, wang1241, wenyang001}@e.ntu.edu.sg,
{kejun.wu, ekhyap}@ntu.edu.sg,   lap-pui.chau@polyu.edu.hk

## A   Dataset Construction and Analysis

In this section, we provide a detailed mask generation description, parameter effect analysis, and more visualization of our dataset.

### A.1   Mask Generation

In video inpainting and error concealment tasks, it is common to generate mask video sequences for simulating different types of video content loss. In view of this, we describe the decoded corruption regions of our dataset by specialized and sophisticate masks. These masks are generated through difference and morphological operations during the model training, evaluation, and testing processes. The flowchart of mask generation in our dataset are shown in Fig. 1.

In Fig. 1, a corrupted frames and a corruption-free frame are presented in the left, which serve as the input of our mask generation processing. Specifically, we calculate the absolute difference between the corruption-free and corrupted frames to obtain the difference maps in three color channels of RGB. In each difference map, the pixel values indicate the magnitude of the difference. By performing histogram normalization, we can improve the contrast of difference maps and then merge them into a single image. To simplify calculations, we convert the RGB difference map to grayscale domain. Then, we perform binarization to grayscale maps. Pixel values range from 20 to 255 are assigned 255 (white), and other remaining pixel values are set to 0 (black), so that binary maps can be generated. We then define two kernels to perform morphological operations. Kernel 1 is an $8 \times 8$ block, and Kernel 2 is a $3 \times 3$ matrix with an elliptical structuring element. These kernels are applied in opening operations for 3 iterations, dilation for 1 iteration, and closing for 3 iterations. The morphological process can remove noise, separate distinct elements, close small holes, and connect adjacent elements of the binary map. Finally, we detect all enclosed contours in the binary map. These slight enclosed contours occupied less than 500 pixels are discarded to suppress excessive mask regions. The rationality of this operation comes from that slight image demange is hard to be perceptible by humnan eyes. In this way, we can create the masks for all corrupted frames in each video of our dataset.

---

[*]Equal first contribution
[†]Corresponding author
[‡]Yi Wang was with NTU when this research was conducted and is currently with Hong Kong PolyU.

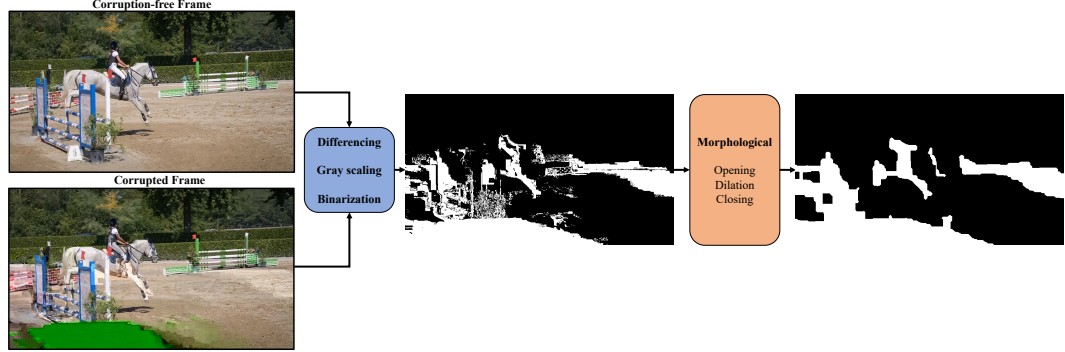

Figure 1: The flowchart of mask generation.

## A.2 Parameter Effects

In the paper, three-parameter corruption model $(P, L, S)$ is introduced construct the dataset from corrupted bitstream. The corrupted fragments are defined by frame corruption probability $P$, corruption location $L$, and fragment size $S$. The model can demonstrate the effect of parameter adjustments on the statistical information of the dataset branch in our paper. Here, we provide a visual representation of this effect. Four parameter settings $(1/16, 0.4, 2048)$, $(1/16, 0.4, 4096)$, $(1/16, 0.2, 4096)$, and $(2/16, 0.4, 4096)$ are adopted. The selected videos' first 48 frames (the lengths of 3 GOPs (group of pictures)) are presented along with their binarized difference maps and the generated masks. Note that the methods for acquiring difference maps and masks are given in Sec. A.1 and Fig. 1.

We take 4 videos from the test set of our dataset as example for illustration. For the 4 videos, the visualizations of decoded frames, difference maps and masks are shown in Fig. 5, Fig. 2, Fig. 3, and Fig. 4. In each Figure, the first top 3 sequences represent the decoded frames, difference maps and masks in the first GOP. The second and third GOPs are also presented below. Generally, the earlier a segment location in a frame bitstream is, the more critical the information it contains. Thus, we can observe from the decoded frame sequence: when reducing corruption location parameter $L$, the corruption location inside a frame shifts ahead and errors occur earlier in a frame. As a result, the areas of damage are larger in a decoded frame, the difference maps and masks are also larger. When increasing the fragment size $S$ of corruption, the decoded frame sequences suffer a severer image quality degradation. This is because more amount of bitstream get lost; thus, larger areas of errors are generated in the frame, which is decoded from the corrupted fragment bitstream. Moreover, the error propagation can be observed among neighboring frames of the corrupted frame, due to reference information missing or being unavailable in inter-frame prediction of video coding. We further increase the corruption probability $P$ inside each GOP, more frames of NALUs (network abstraction layer units) in a bitstream are possibly corrupted; thus, more numbers of corrupted frames are decoded from the bitstream. Due to error propagation among neighboring frames, more amount of corruption can be observed in the decoded frame sequences of each Figure.

From Fig. 5, Fig. 2, Fig. 3, and Fig. 4 we can conclude that: the existence of bitstream corruption may cause varying degrees of decoded frame errors, including isolated corruption within the current frame, short-term corruption to adjacent frames, and long-term corruption throughout the remaining frames of the GOP. Note that the relationship between decoded frame damage and bitstream corruption is not simply linear. Generally, damage regions in a frame has no point-by-point responses to bitstream corruption due to the error propagation. Although it is inaccurate to describe the occurrence of decoded errors as completely random, it can be considered unpredictable.

To summarize, we conduct experiments and analyzes to demonstrate the effects of three-parameter model $(P, L, S)$ parameter settings on dataset construction. We found that the more severer decoded errors will generally occur when adopting an earlier/smaller corruption location $L$, an larger fragment size $S$, and a higher corruption probability $P$.

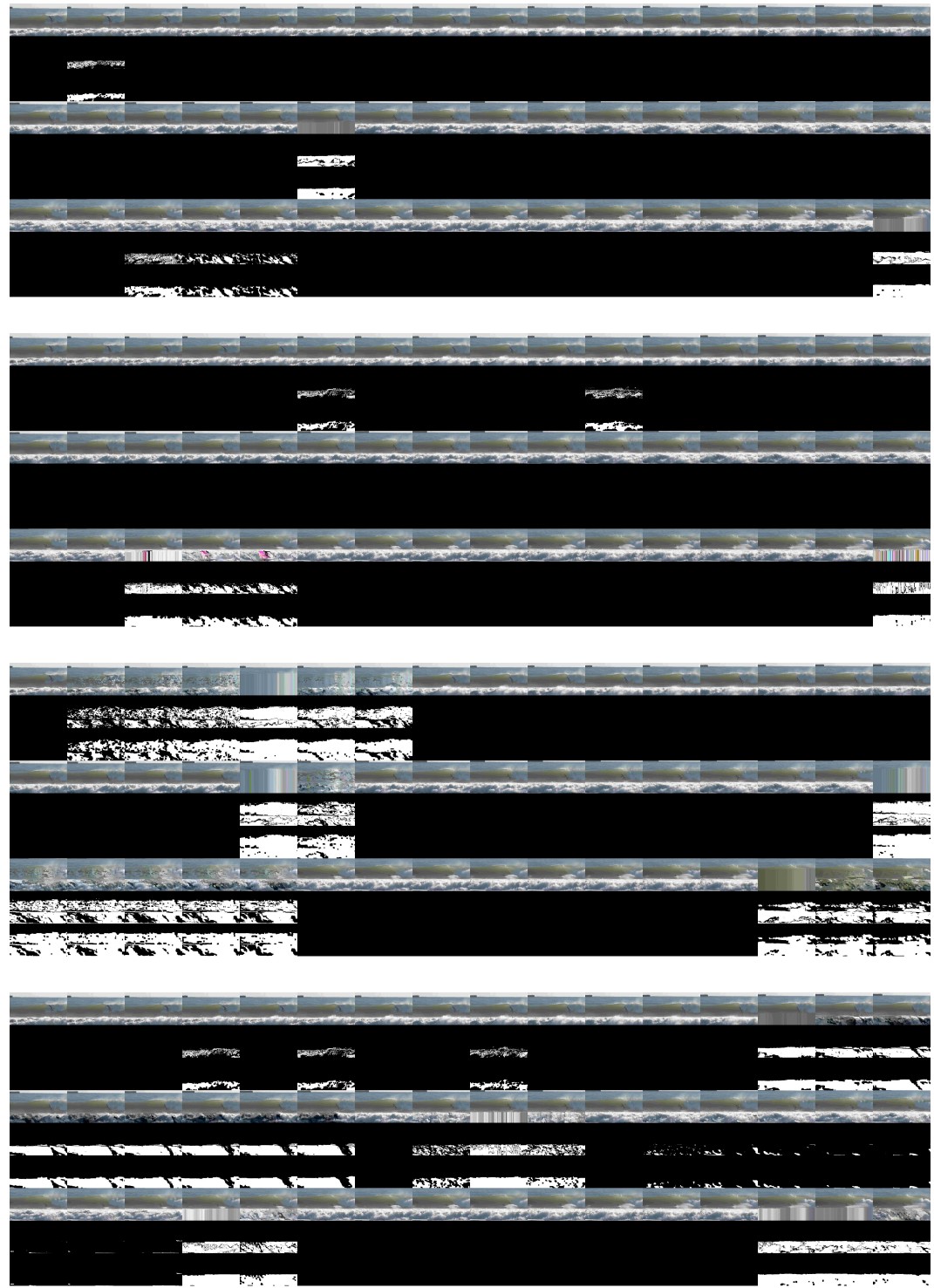

Figure 2: Visualized frame sequence, difference map sequence, and mask sequence of sampled corrupted video in our dataset. From top to bottom, the $(P, L, S)$ parameter settings are $(1/16, 0.4, 2048)$, $(1/16, 0.4, 4096)$, $(1/16, 0.2, 4096)$, $(2/16, 0.4, 4096)$, respectively.

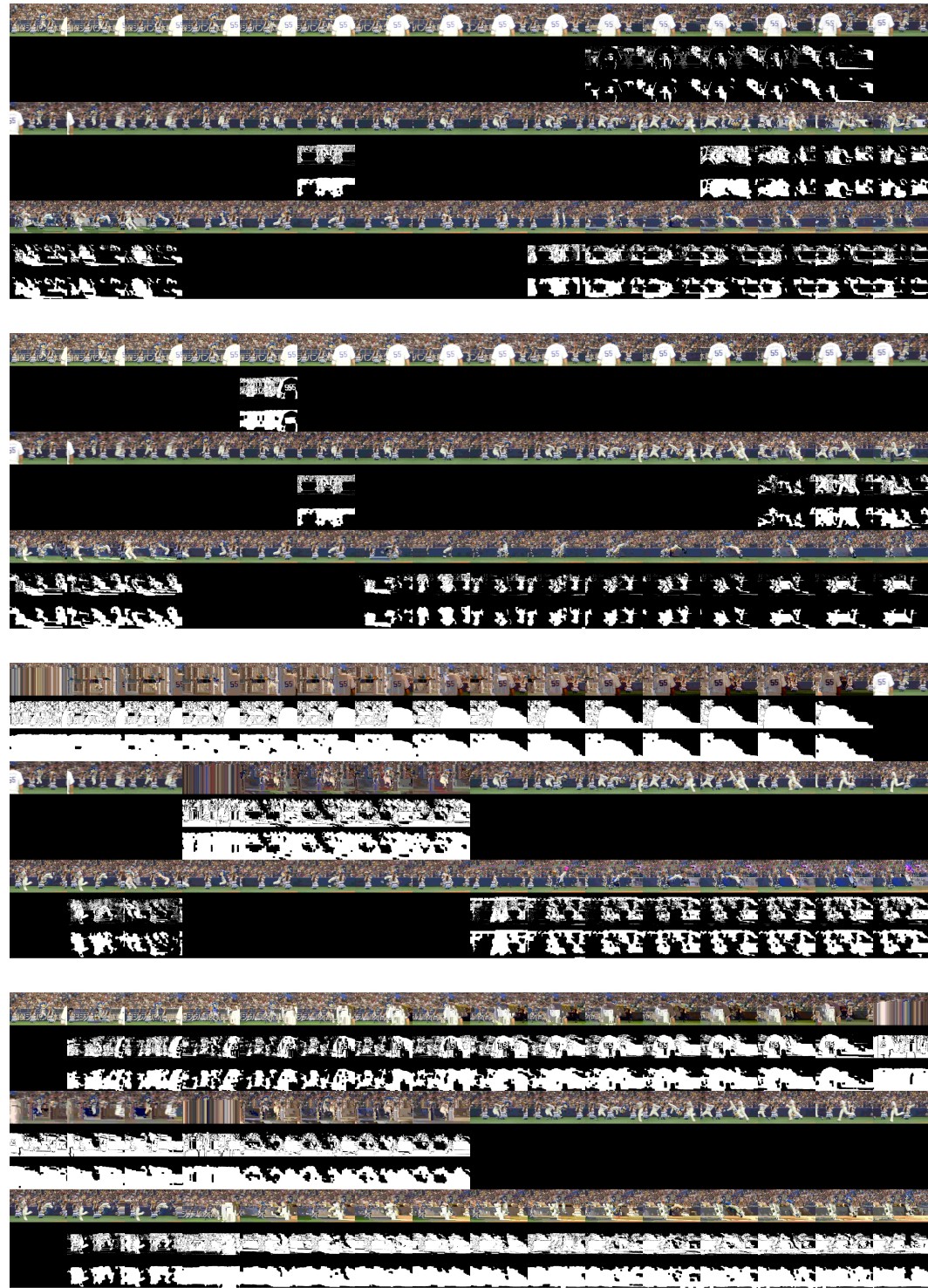

Figure 3: Visualized frame sequence, difference map sequence, and mask sequence of sampled corrupted video in our dataset. From top to bottom, the $(P, L, S)$ parameter settings are $(1/16, 0.4, 2048)$, $(1/16, 0.4, 4096)$, $(1/16, 0.2, 4096)$, $(2/16, 0.4, 4096)$, respectively.

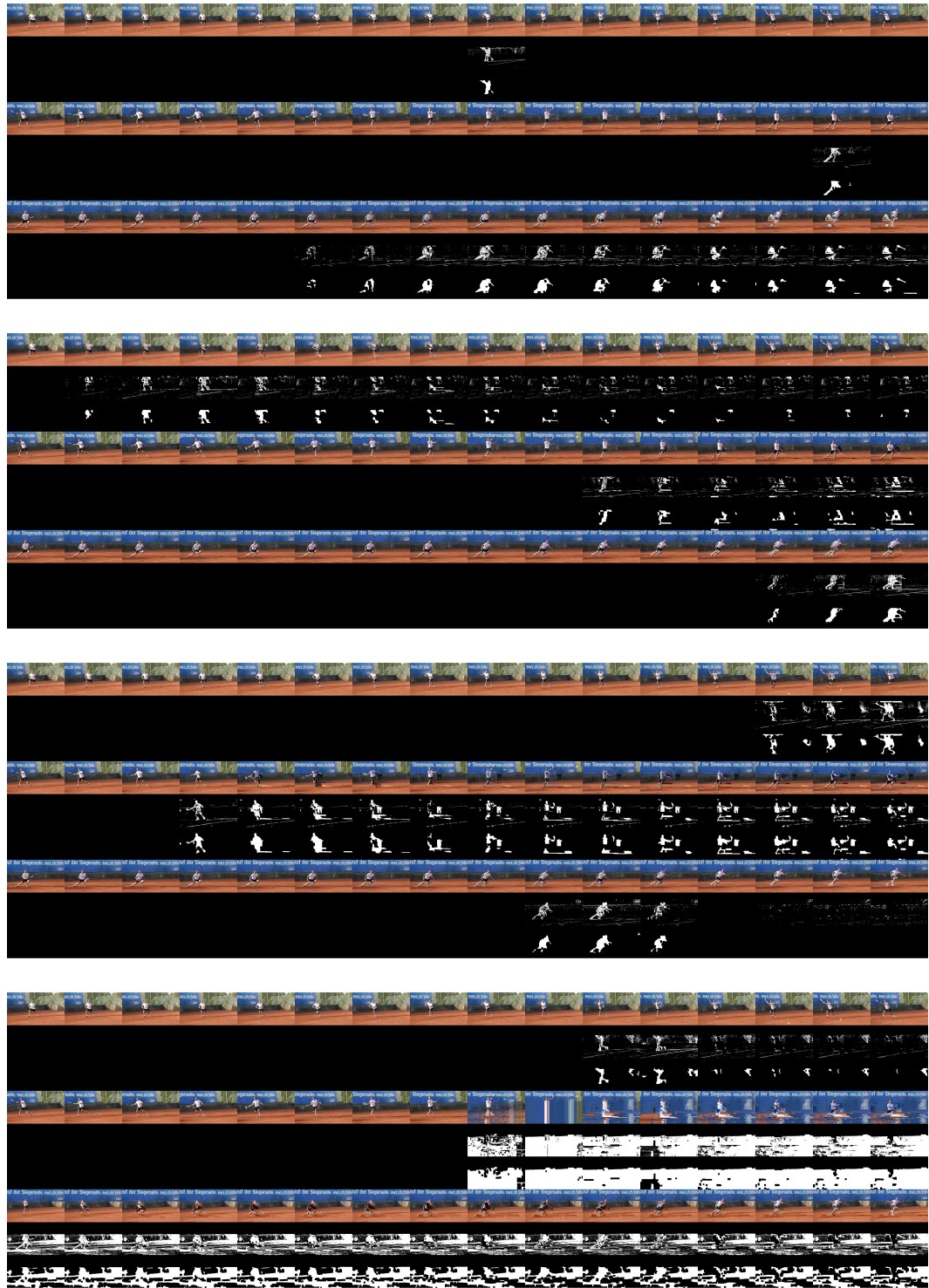

Figure 4: Visualized frame sequence, difference map sequence, and mask sequence of sampled corrupted video in our dataset. From top to bottom, the $(P, L, S)$ parameter settings are $(1/16, 0.4, 2048)$, $(1/16, 0.4, 4096)$, $(1/16, 0.2, 4096)$, $(2/16, 0.4, 4096)$, respectively.

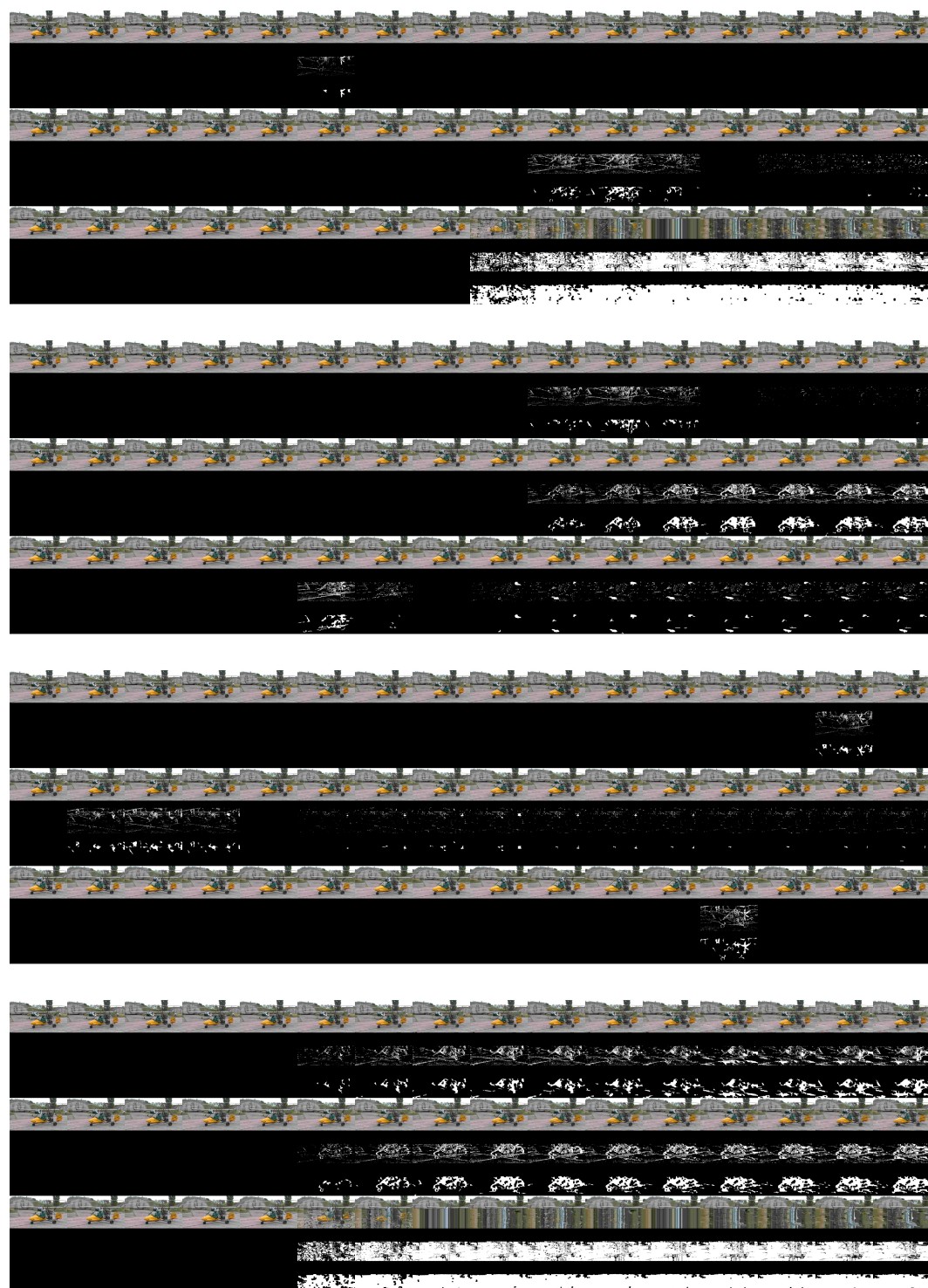

Figure 5: Visualized frame sequence, difference map sequence, and mask sequence of sampled corrupted video in our dataset. From top to bottom, the $(P, L, S)$ parameter settings are $(1/16, 0.4, 2048)$, $(1/16, 0.4, 4096)$, $(1/16, 0.2, 4096)$, $(2/16, 0.4, 4096)$, respectively.

# B    BSCVR Framework Detail

This section provides the details of our bitstream-corrupted video recovery (BSCVR) method inspired by end-to-end video inpainting framework [1]. Our BSCVR is composed of segmented feature extraction and feature completion for better subsequent recovery process. We also introduce content recovery based on [1].

## B.1    Feature Extraction

The commonly used context encoder [2] is applied for corrupted images where it takes an input image with missing regions to extract local temporal neighboring features outside the masked region. In our bitstream corruption scenario, we modified the framework with additional feature extraction in corrupted contents. The corrupted sequence $\{X^t|t = 1, 2, ..., T_l\}$ is separated as a "corruption-free region sequence" $\left\{P^t = (X^t * M^t)^t \in \mathcal{R}^{3 \times h \times w}|t = 1, 2, ..., T_l\right\}$, and a "corrupted region sequence" $\left\{Q^t = (X^t * (M^t)^c)^t \in \mathcal{R}^{3 \times h \times w}|t = 1, 2, ..., T_l\right\}$, where $T_l$ denotes the length of neighboring frames, $M^t$ is the mask of the corrupted region at Frame $t$, and $(M^t)^c$ is the complement of $M$. The context encoder $f(\cdot)$ will be applied to extract the local neighboring features with a downsampling factor of $1/n$,

$$\left\{[E_P^t; E_Q^t] = [f(P^t); f(Q^t)]|t = 1, 2, ..., T_l\right\}, \tag{1}$$

where $E_P^t \in \mathcal{R}^{(C \times \frac{h}{n} \times \frac{w}{n})}$ and $E_Q^t \in \mathcal{R}^{(C \times \frac{h}{n} \times \frac{w}{n})}$ are extracted temporal neighboring features with a channel size of $C$ for corruption-free and corrupted region sequences, respectively.

## B.2    Feature Completion

Traditionally, in the video inpainting task, the extracted feature $E_P^t$ of the corruption-free region is directly applied for content generation. The effectiveness of this process is usually influenced by multiple factors. One is that the content information of the corruption-free regions cannot cooperate well with the motion information indicated by optical flow. This problem causes unreliable propagation using only information provided by local temporal neighbors. Research on video inpainting has already pointed out that imprecise flow estimation and lacking local content usually lead to unsatisfactory content generation, especially in large corrupted regions.

Based on the observation of bitstream-corrupted videos, we found that most decoding results in the corrupted regions can somewhat preserve key visual information such as color, texture, and object contours. Although the regions are distorted visually, they can provide useful information in the feature space. Therefore, we design a module based on the self-attention mechanism [3], which aims to complete and enhance the representative quality of features leveraging the information in the corrupted region.

Given two region-based features $E_P^t$ and $E_Q^t$ of the $t$-th frame, we first concatenate them and generate the feature $\left\{E_{PQ}^t \in \mathcal{R}^{(2C \times \frac{h}{n} \times \frac{w}{n})}|t = 1, 2, ..., T_l\right\}$. By applying 1x1 convolution, we perform dimension reduction to facilitate better processing. The features after dimensionality reduction

Figure 6: Detailed structure of the proposed feature completion module

$\left\{ E_{PQ_\downarrow}^t \in \mathcal{R}^{(\frac{C}{2} \times \frac{h}{n} \times \frac{w}{n})} | t = 1, 2, ..., T_l \right\}$ is then fed into the Transformer encoder. The encoder consists of $L_{enc}$ layers, each containing a multi-head self-attention module followed by a feed-forward neural network and Layer normalization. The output of is a sequence of encoded patches $\mathbf{P} \in \mathbb{R}^{F \times \frac{h}{n} \times \frac{w}{n}}$, where $F$ is the intermediate feature size. This step can achieve global information interaction and non-local dependency modeling of image features. Next, the encoded patches $\mathbf{P}$ are used to generate the completed feature by a Transformer decoder consists of $L_{dec}$ layers, which restores the self-attention map to the original feature map. These steps can achieve feature refinement and retain the advantages of global information interaction and non-local dependency modeling established in the Transformer encoder. After the refinement process, the patches are rearranged back into a feature map $E_{PQ_\downarrow}^{'t}$ using the inverse operation of the initial patch extraction. The refined feature map then passes through another several convolutional layers $\mathcal{C}(\cdot)$, which fuses the two content information in a concatenated feature vector. Consequently, with channel fusion, skip connection, and output projection, the representative capability of the resulting feature can be further enhanced by fusing multi-scale and multi-level information as we discussed in main paper. The obtained feature can be denoted as

$$\left\{ E^{'t} = \mathcal{C}(E_{PQ_\downarrow}^{'t}) \in \mathcal{R}^{(C \times \frac{h}{n} \times \frac{w}{n}} | t = 1, 2, ..., T_l \right\}. \tag{2}$$

With the enhanced expression ability and restoration quality of the features, the content generation process will be beneficial, making the model performer better in the bitstream-corrupted video recovery problem.

## B.3 Content Recovery

In this section, we briefly introduce the modules originally used in video inpainting framework, i.e., the E2FGVI [1].

**Flow Completion.** In severely corrupted videos, directly generating repaired image content is challenging or impossible. Therefore, a commonly used method is to complete the video's optical flow instead of directly generating the content [4]. Following existing flow-guided methods, firstly, a learnable flow completion module based on SPyNet [5] is used to estimate the forward and backward optical flow of the down-scaled frame sequence with erased, corrupted regions and the loss for the forward and backward flow restoration. The ground truth flow is obtained by loading pre-trained SPyNet and predicting the optical flow of the undamaged frame sequence.

**Flow-guided feature propagation.** Taking the completed local temporal neighboring features and predicted forward and backward flows of the corrupted frame, the motion in the corrupted region can be estimated to guide the content generation. With feature propagation proposed by [1] using modulated deformable convolution [6], the candidate feature points for inaccurate flow estimation can be better rearranged and therefore provide better sampling position for the reference content searching.

**Content generation.** With the fused propagation features, we can construct the overall feature of the local frames by concatenating and soft spilting [7] the propagation feature and the context encoder-extracted reference feature from randomly sampled non-local frames. Finally, temporal focal transformer [1] capable of efficient computation with window-based attention [8, 9] and adaptive grain attention is adopted to process sequence data rather than original focal transformer [8]. We can generate the recovery result by a decoder module.

## B.4 Training method.

Inspired by existing video inpainting methods, we apply multiple loss terms from different perspectives [1] to our BSCVR framework. The overall loss function is composed of three terms $\mathcal{L}_{rec}$, $\mathcal{L}_{adv}$, and $\mathcal{L}_{flow}$, expressed as:

$$Loss = \mathcal{L}_{rec} + \mathcal{L}_{adv} + \mathcal{L}_{flow} = \left\| \hat{\mathbf{Y}} - \mathbf{Y} \right\|_1 + (-E_{z \sim P_{\hat{\mathbf{Y}}}(z)}[D(z)]) + \mathcal{L}_{flow}. \tag{3}$$

Meanwhile, since the model is trained in GAN [10] paradigm, the applied T-PatchGAN [11] based discriminator should minimize the loss, expressed as:

$$\mathcal{L}_{\mathcal{D}} = E_{x \sim P_Y(x)}[\text{ReLU}(1 - D(x))] + E_{x \sim P_{\hat{Y}}(x)}[\text{ReLU}(1 + D(z))]. \tag{4}$$

## C  Supplementary Experiments

### C.1  Implementation Setting of BSCVR

The implemented feature completion module consist of 2 layers of Transformer encoder and 1 layer of Transformer decoder to perform feature re-coding and re-generation in our experiment. The feature map's height and width is down scaled by factor $n = 4$ for computational efficiency in our experiments.

During training, following the existing works [12, 7, 1], the weights of $\mathcal{L}_{rec}$, $\mathcal{L}_{adv}$, and $\mathcal{L}_{flow}$ are $1$, $10^{-2}$, and $1$, respectively. Considering GPU memory limits, we trained the model with resized $432 \times 240$ frames. Local frames were obtained by some continuous clips, and non-local frames were randomly sampled from the videos. The number of local frames ($T_l$) and non-local frames ($T_{nl}$) was set to 5 and 3, respectively.

Following STTN [12], FuseFormer [1], and E2FGVI [1], during evaluation and test, we used a sliding window with the size of 10 to get local neighboring frames and uniformly sampled the non-local neighboring frames with a sampling rate of 10. We adopted the Adam optimizer with $\beta_1 = 0$ and $\beta_2 = 0.99$. The initial learning rate was set as 0.0001 for all modules and reduced by a factor of 10 at 80% of all iterations. We used 4 NVIDIA 3090 GPUs for training, and the batch size was set to 8, the CPU is AMD Ryzen Threadripper 3990X 64-Core Processor.

### C.2  More Qualitative Results

Here we provide additional qualitative results to show the superiority of the proposed BSCVR. The reconstruction results of baselines are also presented for comparison. Figs. 7 to 8 show that our BSCVR can generate more faithful textural and structural information and more coherent contents in masked regions than other methods.

#### C.2.1  Performance Stability

For applying existing video inpainting methods to bitstream-corrupted video recovery, one common failure case is the continuous and large masks, such that the recovery result is not faithful, and they usually tend to remove the corrupted objects in the frame. This tendency is stronger when the frame is far away from its closest corruption-free frames. As Figs. 10 to 12 show, we further provide some video clips with continuous and relatively big masks for demonstrating the performance stability of our methods.

#### C.2.2  Comparison with SOTA non-end-to-end method

We further compared our method with latest non-end-to-end flow-based video inpainting methods. Although non-end-to-end methods theoretically have their drawbacks, their independent optical flow completion module usually results in better inpainting performance in practice. Researchers are constantly exploring and comparing the performance of these two designs, but there is actually no complete conclusion yet. However, it is currently clear that independent optical flow completion modules require independent computing resources for training and inference, as well as the large amount of CPU operations they contain, which yields disadvantage in efficiency in practical application.

We selected the SOTA non-end-to-end method ECFVI [13] and FGT [14] for comparison. It should be noted that the pretained model of FGT in comparison requires significantly more training iterations with 280K iterations for flow completion model training and 500K for inpainting model. The quantitative evaluation result is shown in Tab. 1. The qualitative evaluation is shown in Fig. 13

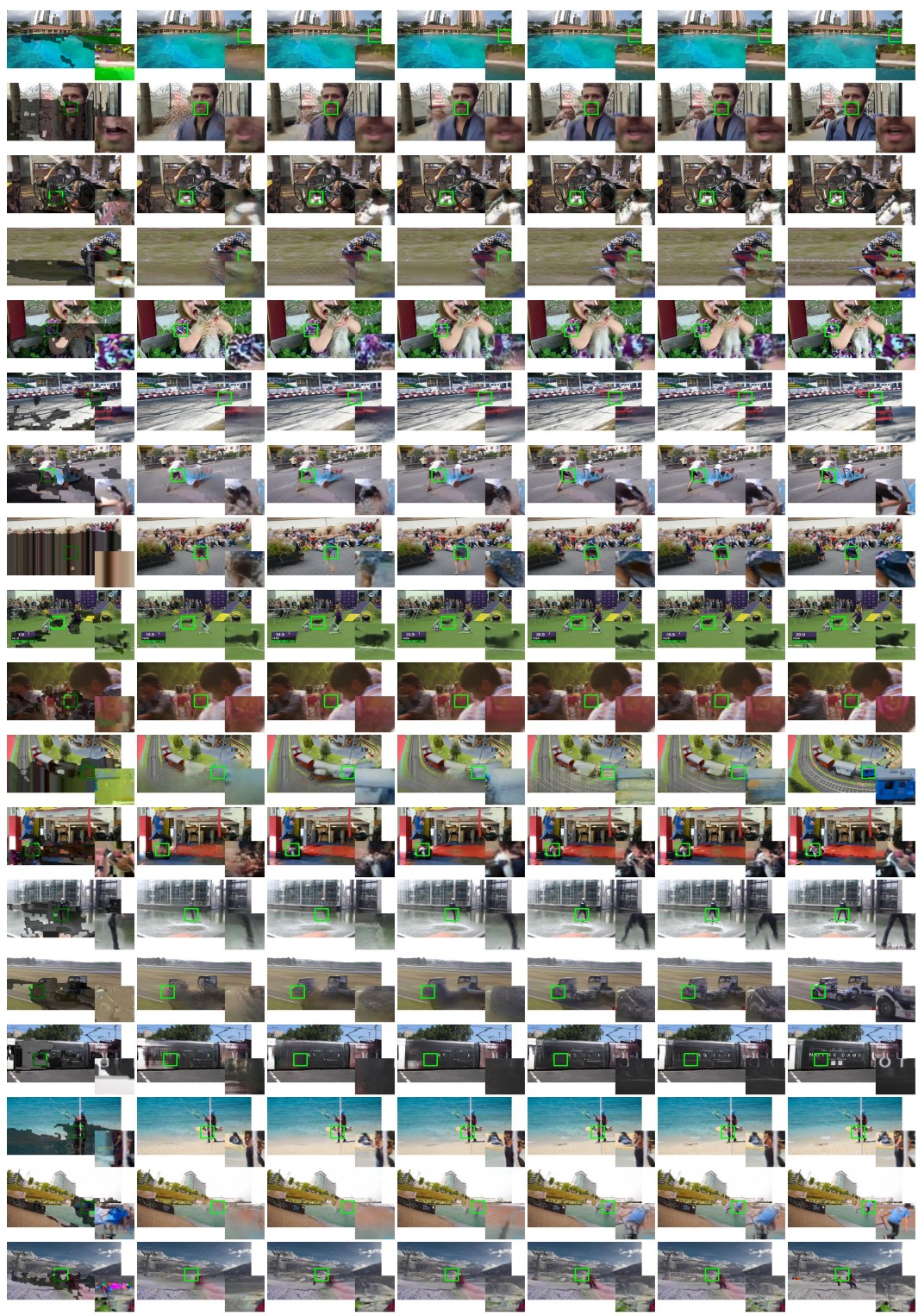

Figure 7: Visual comparison in 240P resolution, the order from left to right is the same as in the main paper: in sequence, they are input, STTN [12], FuseFormer [7], E2FGVI [1], BSCVR-S, BSCVR-P, GroundTruth.

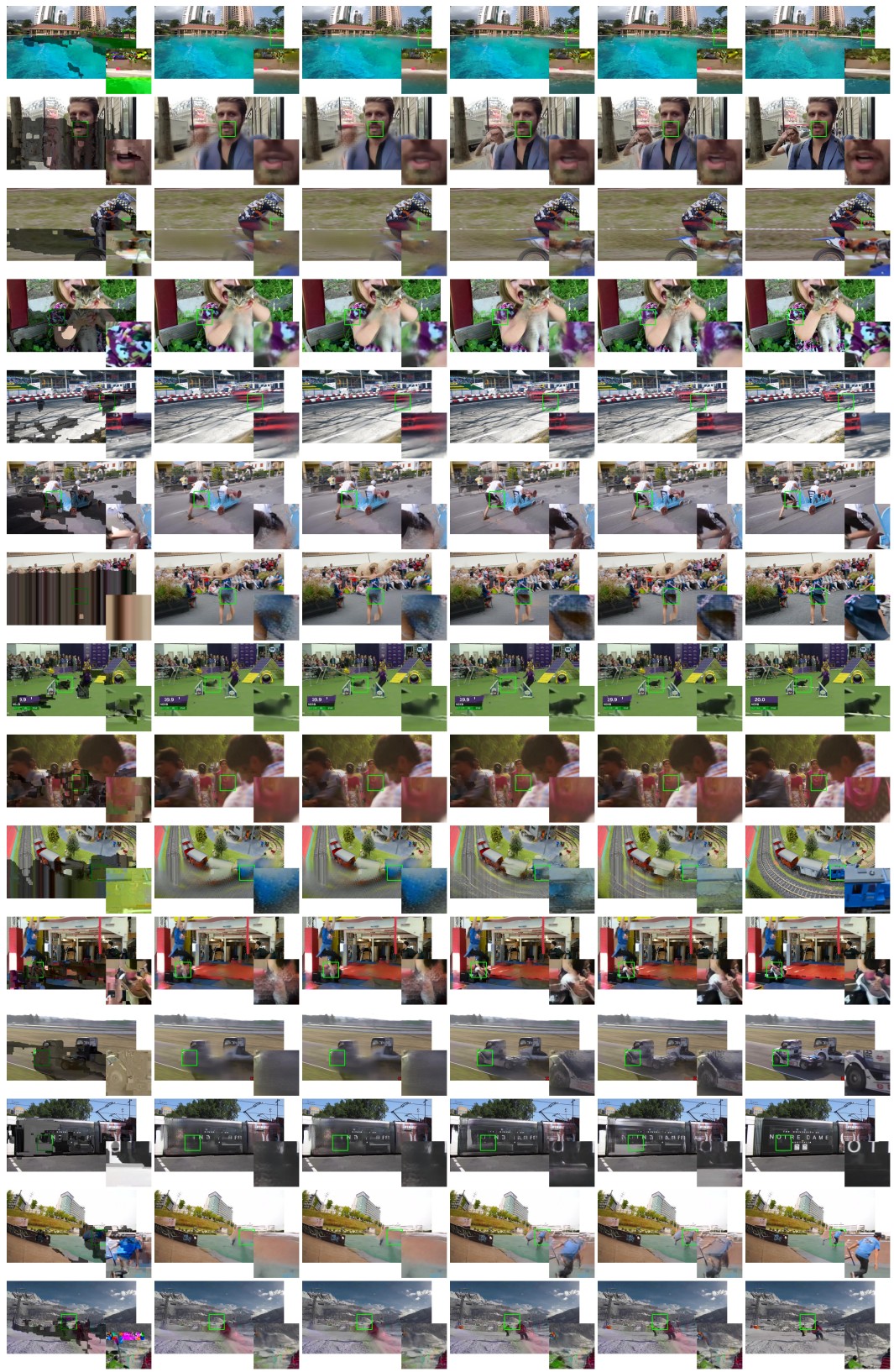

Figure 8: Visual comparison in original resolution, the order from left to right is the same as in the main paper, in sequence, they are input, pretrained E2FGVI [1], bsc-trained E2FGVI [1], BSCVR-S, BSCVR-P, GroundTruth.

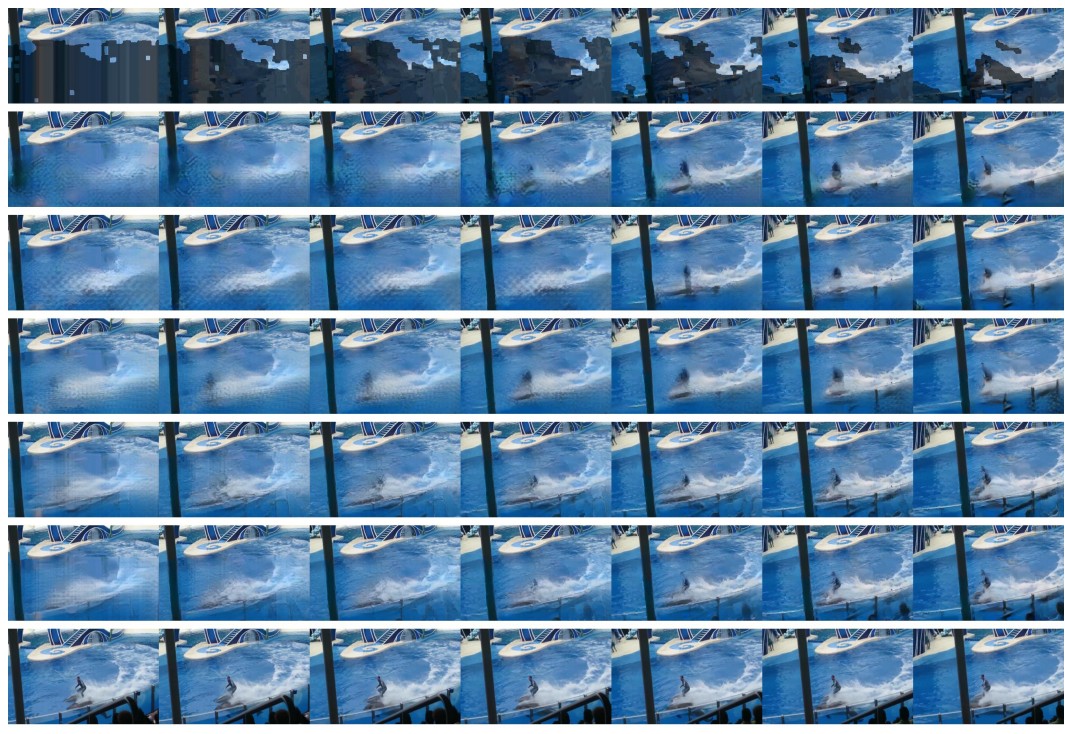

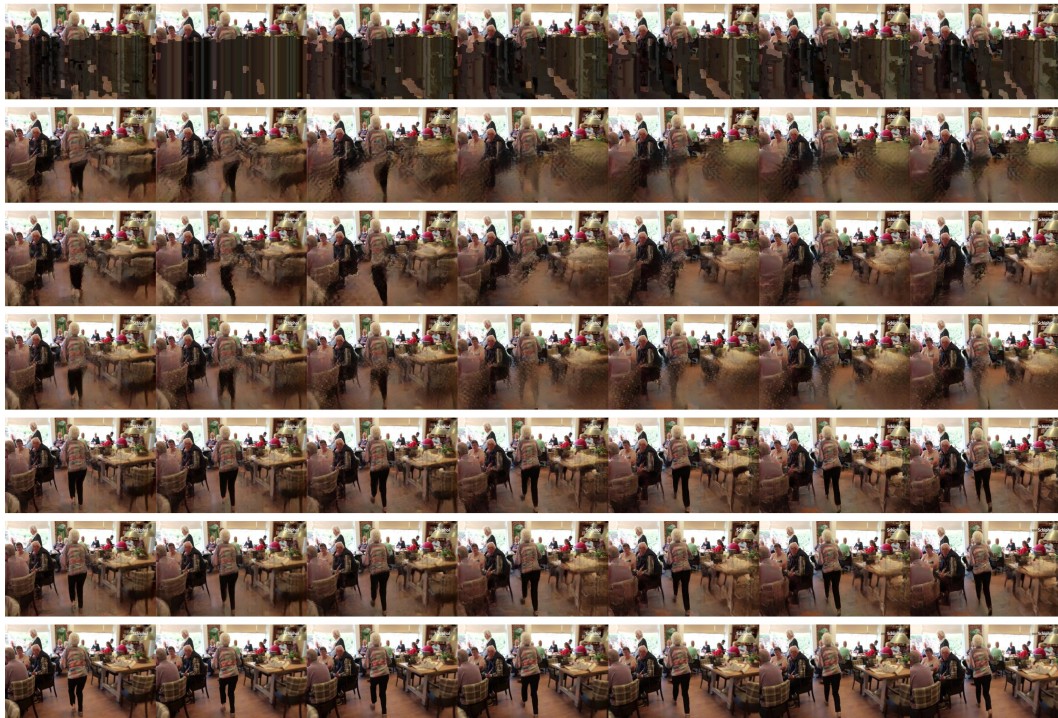

Figure 9: Performance stability visualization in 240P resolution. For each visualized clip, from top to bottom is the input clip, STTN, FuseFormer, E2FGVI, BSCVR-S, BSCVR-P, GroundTruth, in sequence

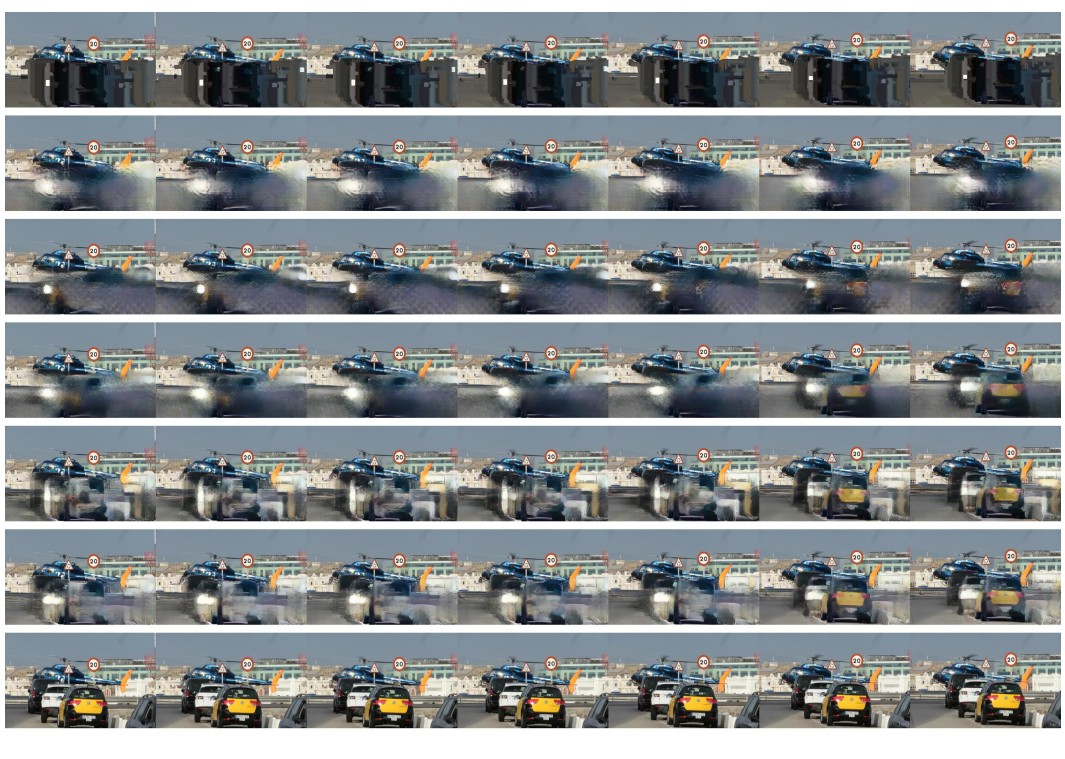

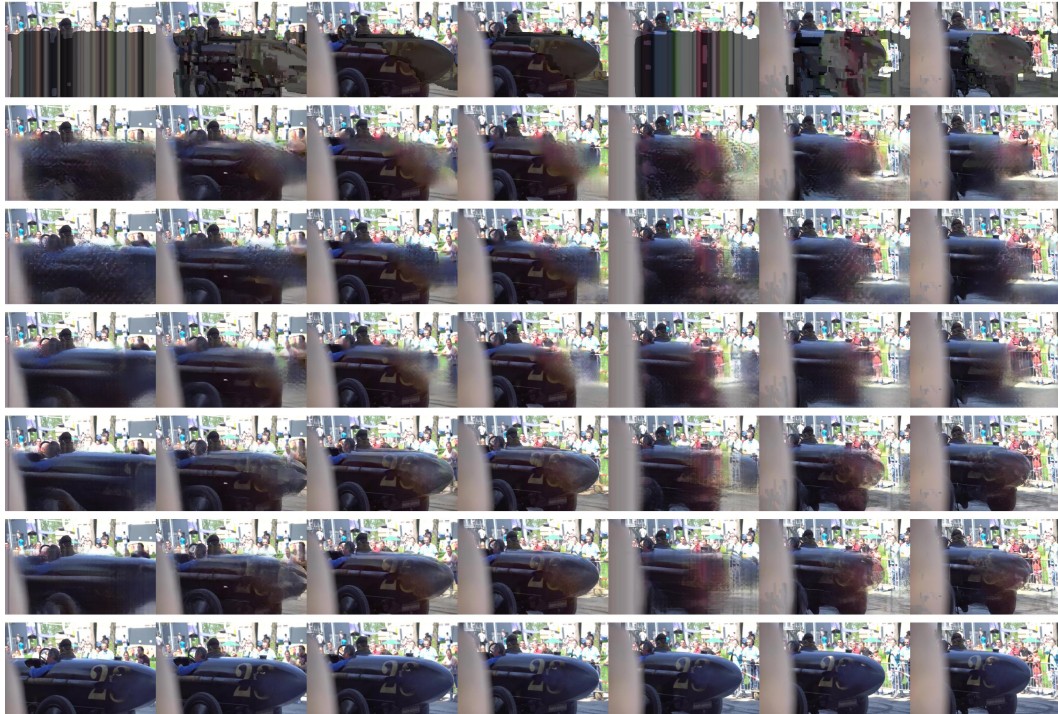

Figure 10: Performance stability visualization in 240P resolution. For each visualized clip, from top to bottom is the input clip, STTN, FuseFormer, E2FGVI, BSCVR-S, BSCVR-P, GroundTruth, in sequence

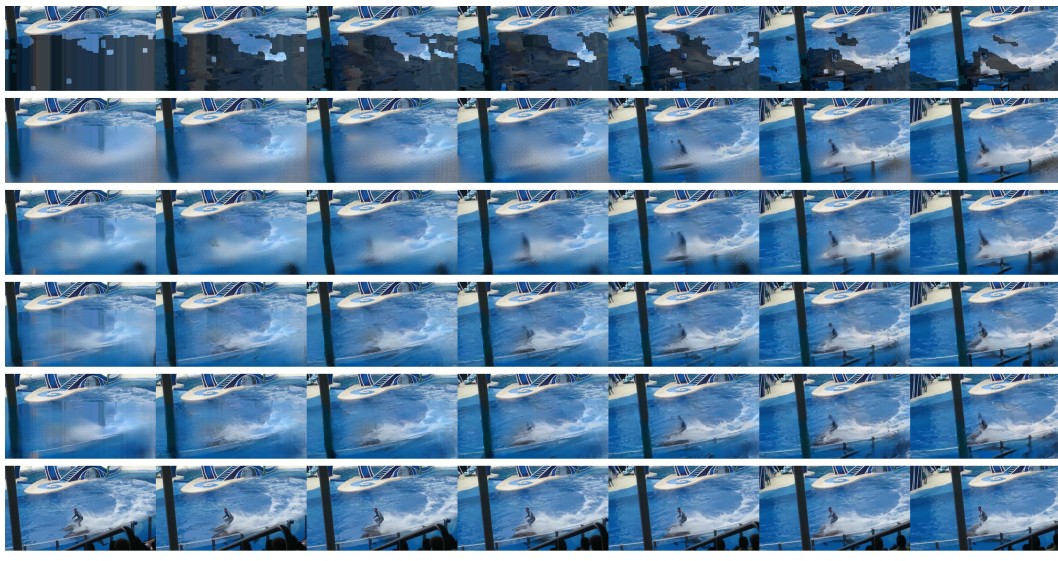

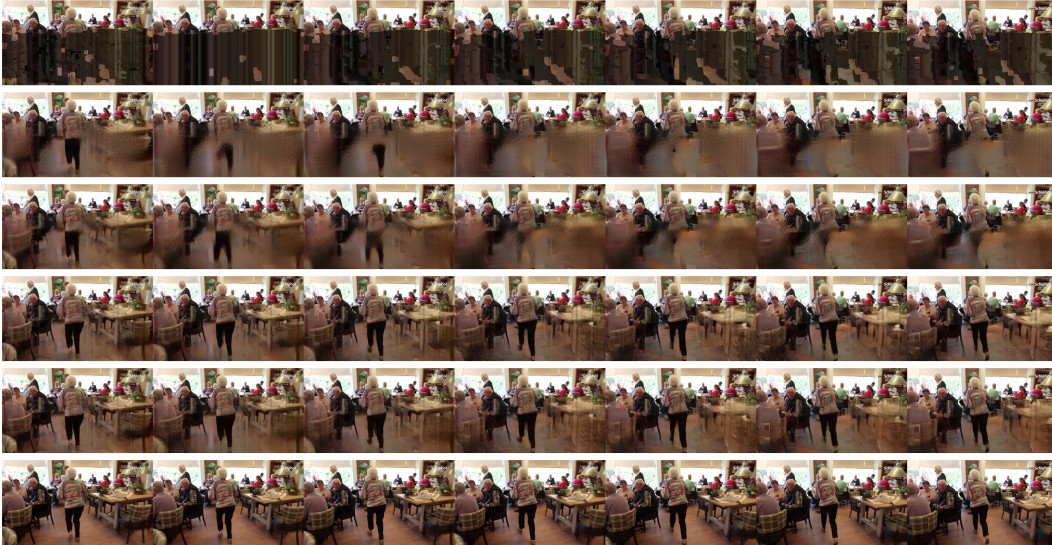

Figure 11: Performance stability visualization in original resolution. For each visualized clip, from top to bottom is the input clip, pretrained E2FGVI, BSC-trained E2FGVI, BSCVR-S, BSCVR-P, GroundTruth, in sequence

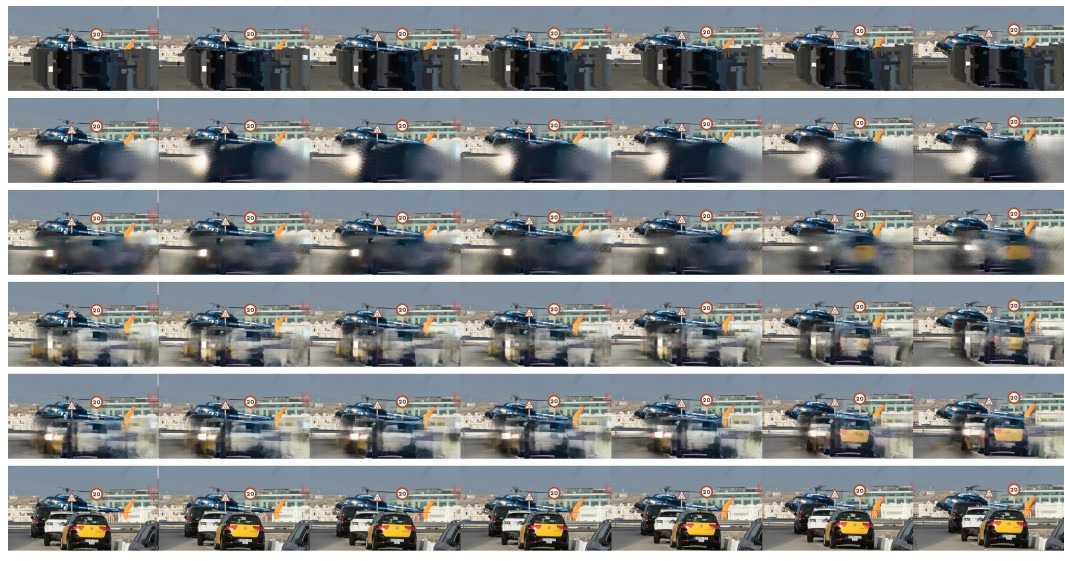

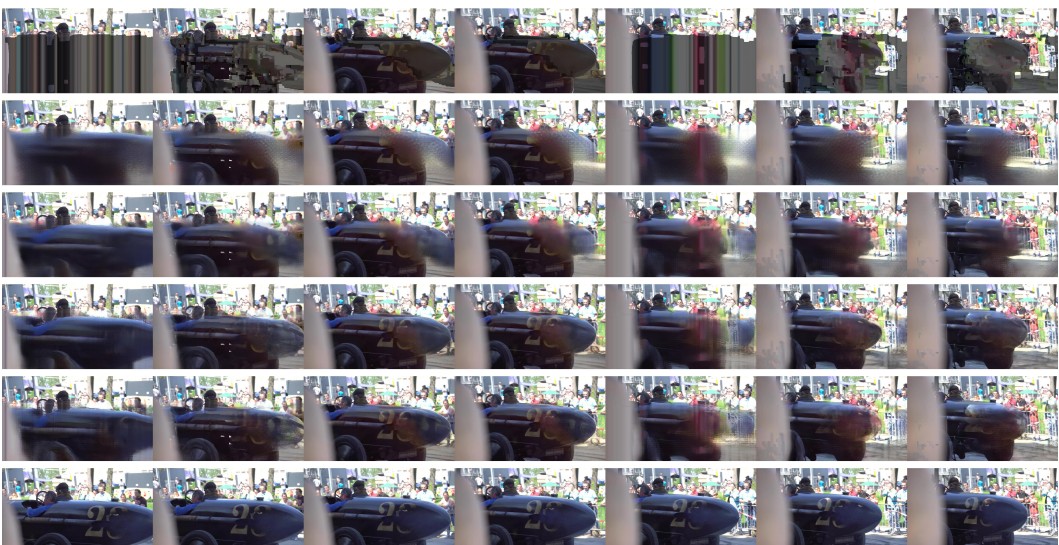

Figure 12: Performance stability visualization in original resolution. For each visualized clip, from top to bottom is the input clip, pretrained E2FGVI, BSC-trained E2FGVI, BSCVR-S, BSCVR-P, GroundTruth, in sequence

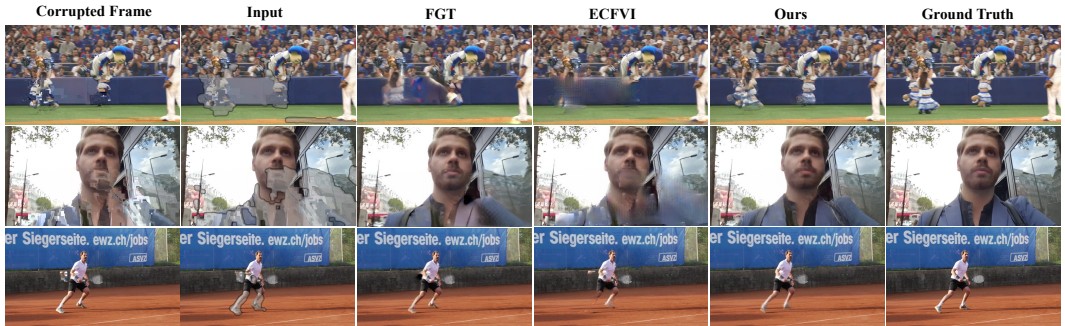

| Corrupted Frame | Input | FGT | ECFVI | Ours | Ground Truth |

Figure 13: Qualitative evaluation with pretrained SOTA non-end-to-end methods under 240P resotion.

| Test res. | Method | Accuracy | | | | | | | | Efficiency |
| | | YouTube-VOS (720P) Subset | | | | DAVIS (480P) Subset | | | | Runtime |
| | | PSNR↑ | SSIM↑ | LPIPS↓ | VFID↓ | PSNR↑ | SSIM↑ | LPIPS↓ | VFID↓ | s/frame |
| 240P | Input | 18.8749 | 0.8160 | 0.1527 | 0.2015 | 18.4562 | 0.7921 | 0.1541 | 0.4189 | - |
| | ECFVI [13] | 22.6254 | 0.8524 | 0.0556 | 0.0981 | 20.6013 | 0.7758 | 0.0678 | 0.2219 | ∼ 2.24 |
| | FGT [14] | **33.5220** | 0.9288 | 0.0421 | 0.1289 | **32.3549** | 0.9121 | 0.0421 | 0.2380 | ∼ 1.97 |
| | **BSCVR-S (Ours)\*** | 31.8345 | 0.9584 | 0.0262 | 0.0427 | 28.4211 | 0.9180 | 0.0381 | 0.1196 | 0.172 |
| | **BSCVR-P (Ours)\*** | 31.9534 | **0.9598** | **0.0258** | **0.0426** | 28.5430 | **0.9199** | **0.0375** | **0.1165** | 0.178 |

Table 1: Quantitative comparison with pretrained SOTA non-end-to-end method under 240P input resolution.

# Acknowledgement

This research is supported by the National Research Foundation, Singapore, and Cyber Security Agency of Singapore under its National Cybersecurity Research & Development Programme (Cyber-Hardware Forensic & Assurance Evaluation R&D Programme <NRF2018NCRNCR009-0001>). Any opinions, findings and conclusions or recommendations expressed in this material are those of the author(s) and do not reflect the view of National Research Foundation, Singapore and Cyber Security Agency of Singapore.