# OpenReview forum: "Bitstream-Corrupted Video Recovery: A Novel Benchmark Dataset and Method"
_NeurIPS.cc/2023/Track/Datasets_and_Benchmarks — NeurIPS 2023 Datasets and Benchmarks Poster_

### Official Review · Reviewer_rrQM · 2023-06-26
**The main problem is the proposed benchmark might not be interesting for the AI research community.**

**Rating:** 5
**Confidence:** 4
**Correctness:** The claims, and the designs of the pr…
**Clarity:** The presentations are clear.

**Strengths:**

1. It is the first large scale bitstream-corrupted benchmark for the video recovery task.

2. Data, model, and related code are released.

3. The presentations are clear.

**Additional Feedback:**

No additional comments. Please see the above parts.

**Documentation:**

The code and data are publicly available.  It seems to the reviewer that the paper contains sufficient details for reproducing.

**Ethics:**

Since the benchmark is built on the public dataset. There are no ethical concerns.

**Limitations:**

The proposed dataset is for the task of video recovery, which is an important realistic application. But it might not be interesting for the AI research community. Modeling multi-model information in the bitstream directly is an important topic for the foundation models to understand the real world. Consideration in such direction and provide related benchmarks will benefit the developing of foundation models.

**Opportunities For Improvement:**

1. The target application area is not the main focus of the NeurIPS. The proposed benchmark might be more suitable for other conferences.

**Relation To Prior Work:**

The related works are included and discussed.

**Summary And Contributions:**

The authors proposed a bitstream-corrupted video benchmark (BSCV), for the task of video recovery. The BSCV contains 16000 public available video clips. The author also proposed a novel corruption model, which contains 3 parameters, performing on the bitstream. The corruption model is also released.

---

> ### Author Response · Authors · 2023-08-22
> **Response to reviewer rrQM**
>
> Thank you for your comment on our work. Here we would like to discuss with you the issues you mentioned.
>
>
>
> ### Q: The target application area is not the main focus of the NeurIPS. The proposed benchmark might be more suitable for other conferences.
>
> A: (1) “Vision” is one of the fundamental areas of NeurIPS. AI-based low-level vision, e.g., image/video restoration, compression, etc. are common topics in both Main Track and Dataset & Benchmark Track of NeurIPS.
>
> (2) Existing AI-based video recovery methods are trained in videos with simulated corruption such manual-designed error masks, block and slices content loss [6-11].
>
> (3) However, real-world video corruption generally happens when bitstream is corrupted in most scenarios, e.g., living stream (TikTok), video conference (Zoom), internet videos (YouTube) and other video communications.
>
> Therefore, AI cannot learn real-world knowledge from existing simulated datasets.
>
> (4) Our work is the first large-scale bitstream-corrupted benchmark to drive the AI to learn the real-world knowledge for video recovery. We believe our work can facilitate the AI-based vision research to solve the real-world problem
>
> (5) We would like to share some related works [1-5] published in NeurIPS in the last two years. These works made great contributions to the low-level vision research.
>
> **Reference**
>
> [1] Antsiferova et al. "Video compression dataset and benchmark of learning-based video-quality metrics." Advances in Neural Information Processing Systems 35 (2022): 13814-13825. **(Datasets and Benchmarks Track)**
>
> [2] Aakerberg, Andreas, Kamal Nasrollahi, and Thomas B. Moeslund. "RELLISUR: a real low-light image super-resolution dataset." Thirty-fifth Conference on Neural Information Processing Systems **Datasets and Benchmarks Track (Round 2)**. 2021.
>
> [3] Liang, Jingyun, et al. "Recurrent video restoration transformer with guided deformable attention." **Advances in Neural Information Processing Systems 35 (2022)**: 378-393.
>
> [4] Zhou, Shangchen, et al. "Towards robust blind face restoration with codebook lookup transformer." **Advances in Neural Information Processing Systems 35 (2022)**: 30599-30611.
>
> [5] Fu, Xueyang, et al. "Unfolding Taylor's Approximations for Image Restoration." **Advances in Neural Information Processing Systems 34 (2021)**: 18997-19009.
>
> [6] Yanhong Zeng, Jianlong Fu, and Hongyang Chao. Learning joint spatial-temporal transformations for video inpainting. In Computer Vision–ECCV 2020: 16th European Conference, Glasgow, UK, August 23–28, 2020, Proceedings, Part XVI 16, pages 528–543. Springer, 2020
>
> [7] Rui Liu, Hanming Deng, Yangyi Huang, Xiaoyu Shi, Lewei Lu, Wenxiu Sun, Xiaogang Wang, Jifeng Dai, and Hongsheng Li. Fuseformer: Fusing fine-grained information in transformers for video inpainting. In Proceedings of the IEEE/CVF International Conference on Computer Vision, pages 14040–14049, 2021.
>
> [8] Zhen Li, Cheng-Ze Lu, Jianhua Qin, Chun-Le Guo, and Ming-Ming Cheng. Towards an end-to-end framework for flow-guided video inpainting. In Proceedings of the IEEE/CVF Conference on Computer Vision and Pattern Recognition, pages 17562–17571, 2022.430
>
> [9] Gao, Chen, et al. "Flow-edge guided video completion." Computer Vision–ECCV 2020: 16th European Conference, Glasgow, UK, August 23–28, 2020, Proceedings, Part XII 16. Springer International Publishing, 2020.
>
> [10] Zou, Xueyan, et al. "Progressive temporal feature alignment network for video inpainting." Proceedings of the IEEE/CVF Conference on Computer Vision and Pattern Recognition. 2021.
>
> [11] Xu, Rui, et al. "Deep flow-guided video inpainting." Proceedings of the IEEE/CVF Conference on Computer Vision and Pattern Recognition. 2019.

---

> > ### Author Response · Authors · 2023-08-28
> > **Further response to reviewer rrQM**
> >
> > ### Q: Modeling multi-model information in the bitstream directly is an important topic for the foundation models to understand the real world.
> >
> > 1. To solve video bitstream corruption /packet loss, bitstream processing and vision processing techniques can help enhance the visual quality. Our BSCV (Bitstream-corrupted Video) dataset stems from **bitstream** domain, our recovery method makes full use of the advances of AI in **vision** domain.
> >
> > 2. Studying bitstream directly for video recovery is also an important topic. However, the **application value** of this paper can be regarded as the **last resort** to improve the video quality in video communications. The reasons are as follows:
> >
> >     2.1 Lost data can be retransmitted in some bidirectional channels with feedback, but this approach is not always feasible due to the inherent delay or lacking feedback in unidirectional channels [1].
> >
> >     2.2. Channel coding is another technique by which the data are protected using Forward Error Correction (FEC) codes. Source coding with/without channel coding is known as Error Resilient Coding (ERC). It exploits source coding tools to make the compressed video more resilient against channel errors/losses [2, 3]. For example, multi-stream encoding such as layered video coding or multiple description coding are ERC approaches [4, 5], as well as their combinations [6].
> >
> >     2.3 Although FEC and ERC coded videos can resist errors and losses to some extent, there are scenarios where the received video is too erroneous and still needs further processing [1].
> >
> >     2.4 Due to the predictive nature of the video codecs, the concealment distortion is easily propagated to the future frames which sometime make the video totally unusable [1]. In other word, even though bitstream can get mostly recovered, but the decoded video errors can still be severe.
> >
> >     2.5 In the practical applications, video is captured-> compressed (source coding) -> transmitted (channel coding) -> display (vision domain). Therefore, concealing errors, e.g., by video inpainting, and error concealment can be regarded as the last resort to improve the video quality in case channel coding and source coding are not much effective.
> >
> > 3. The **contributions** of this paper are as follows:
> >
> >     3.1 For existing recovery video methods by vision processing, error/mask patterns, e.g., fixed blocks, slices, or manual-designed masks are remarkably different from the practical damaged video bitstream, illustrated as the redrawn Fig. 1. of the paper.
> >
> >     3.2 To rethink the gap between existing restoration/inpainting methods and practical video corruption characteristics, we constructed the first bitstream corrupted video (BSCV) benchmark dataset and developed method to improve quality from practical video corruption.
> >
> > 4. The **value-add** of this paper to the real world is as follows:
> >
> >     4.1 Video recovery in vision domain no need channels of bidirectional communications, and this approach is more intuitive for video users.
> >
> >     4.2 Our constructed benchmark dataset can reflect the practical video characteristics in video communications. Our method can help enhance end users’ visual fidelity by recovering practical corrupted videos.
> >
> >     4.3 Therefore, this paper has great application value and prospect for most of the popular video scenarios, e.g., internet videos (YouTube, NetFlix), living stream (TikTok), and video conference (Zoom), and other video communications, where bitstream is possible corrupted.
> >
> > 5. In our final camera-ready paper, we will highlight the application value of and advantages of our **bitstream** domain dataset and **vision** domain recovery method.
> >
> > **Reference**
> >
> > [1] Kazemi, M., et, al. (2021). A review of temporal video error concealment techniques and their suitability for HEVC and VVC. Multimedia Tools and Applications, 80, 12685-12730
> >
> > [2] Fleury M, et, al. (2012) Source coding methods for robust wireless video streaming. Multimedia Networking and Coding:175–207
> >
> > [3] Wang Y, et, al. (2000) Review of error resilient coding techniques for realtime video communications. IEEE Signal Processing Magazine 1(4):61–82
> >
> > [4] Kazemi, et, al. (2014). A review of multiple description coding techniques for error-resilient video delivery. Multimedia Systems, 20, 283-309
> >
> > [5] Wang, Y., et, al. (2004). Multiple description coding for video delivery. Proceedings of the IEEE, 93(1), 57-70
> >
> > [6] Ghahremani, S., et, al. (2017). Error resilient video transmission in ad hoc networks using layered and multiple description coding. Multimedia Tools and Applications, 76, 9033-9049

---

### Official Review · Reviewer_BsWN · 2023-07-17
**Bitstream-corrupted Video Recovery: A Novel Benchmark Dataset and Method**

**Rating:** 5
**Confidence:** 4
**Correctness:** Good
**Clarity:** Good

**Strengths:**

+ The proposed task is meanful and interesting.

**Additional Feedback:**

NA

**Documentation:**

Good

**Limitations:**

- The proposed baseline method is too general and the novelty is limited. More advanced video inpainting methods are required to evaluate.
- More quanlitative comparisions are required to clearify the differences between the general  video inpainting and the proposed dataset.

**Opportunities For Improvement:**

- The detail of the baseline method is unclear. In figure 4, why both parts of the correpted frames are as the inputs?

**Relation To Prior Work:**

No.
- More quanlitative comparisions are required to clearify the differences between the general  video inpainting and the proposed dataset.
- More advanced video inpainting methods are required to evaluate.

**Summary And Contributions:**

This paper constructs the first large-scale benchmark, BSCV. The BSCV provides a bitstream corruption model, a realistic decoded video dataset, and a video recovery framework, BSCVR. The bitstream corruption model enables to flexibly generate dataset branches by specifying parameter combinations. The dataset contains 16,000 realistic video clips decoded from corrupted bitstreams with unpredictable error patterns and corruption levels. The BSCVR offers a plug-and-play feature enhancement module to achieve high-quality video recovery. Extensive experiments demonstrate that the proposed BSCVR outperforms SOTA video inpainting methods quantitatively and qualitatively.

---

> ### Author Response · Authors · 2023-08-22
> **Response to reviewer BsWN**
>
> Thank you for your constructive feedback. Here we would like to discuss with you the issues you mentioned.
>
>
>
> ### Q1. The detail of the baseline method is unclear. In figure 4, why both parts of the correpted frames are as the inputs?
>
> A1: Here we redraw Fig.4 and add more explanation to clarify the details of our methods.
>
> (1) **For uncorrupted content input:** Inputting the uncorrupted content is necessary because the encoded local feature is one of the most basic information sources when constructing reference features for video recovery, which is general in prior works [1-5]. We also follow this common practice to form our input channel.
>
> (2) **For corrupted content input:** We additionally enable this input channel because the prior inpainting method ignores the corrupted content and struggles in finding enough guidance when facing corruption in our dataset. This is because we first found that useful information exists in the partial content of the corrupted region, showing the difference between our bitstream corruption video problem and the video inpainting problem. Designing a module with the corrupted content input can help existing video inpainting frameworks expand their perception domain and achieve better recovery feature construction.
>
> ### Q2: The proposed baseline method is too general and the novelty is limited. More advanced video inpainting methods are required to evaluate.
>
> A2: (1) Our novelty includes:
>
> - We first built this large-scale bitstream corrupted video dataset. It differs from previous datasets and provides a realistic task setting. The main feature of this dataset is that it involves complex corruption patterns (error types) and their combinations, which are quite distinct from the simulated corruption patterns in existing datasets.  This fills the gap in this area and is the main contribution for the “Datasets and Benchmarks Track”.
>
> - Aiming to solve this new challenge, our method first enables useful information of the corrupted region as input that was completely ignored by previous frameworks, which is not a general practice.
>
> - Our key idea of firstly adopting transformer architecture for building whole-frame-based local feature is novel for end-to-end framework design. It solves the challenge of constructing global context throughout different parts of the corrupted video.
>
> - Therefore, our work is more effective for the difficult corruptions in real world than previous SOTA inpainting methods.
>
> (2) Our bitstream-corrupted video benchmark presents the video recovery problem from bitstream-corrupted data, treated as a special video restoration problem. We performed a simplification by providing corruption masks so it can be addressed by conventional video inpainting methods. However, the bitstream-corrupted video does not contain those corruption masks. Due to the “Datasets and Benchmarks Track”, we provide such a simplification and the most ideal recovery results by video inpainting methods, and future works should take into account the effective solutions without masks, which is out of the scope of this paper.
>
> Based on the abovementioned issues, we only show some representative video inpainting methods. E2FGVI-HQ is a well-recognized end-to-end video inpainting work published in CVPR2022, and there’s no usable end-to-end method in 2023. And it is the only method could process arbitrary resolution which best matches the demands of our proposed problem. Thus, we omit other baseline methods from further comparison and instead supplement with qualitative evaluations under the original video resolution.
>
>
>
>
>
>
>
> ### Q3: More quanlitative comparisions are required to clearify the differences between the general video inpainting and the proposed dataset.
>
> A3: We conducted a more detailed qualitative evaluation in manuscript, which included richer corruption types and more diverse corruption degrees. Please refer to updated Fig.5 and explanation in line 295 of the paper.

---

> > ### Author Response · Authors · 2023-08-22
> > **Response to reviewer BsWN: Reference**
> >
> > **Reference**
> >
> > [1] Yanhong Zeng, Jianlong Fu, and Hongyang Chao. Learning joint spatial-temporal transformations for video inpainting. In Computer Vision–ECCV 2020: 16th European Conference, Glasgow, UK, August 23–28, 2020, Proceedings, Part XVI 16, pages 528–543. Springer, 2020
> >
> > [2] Rui Liu, Hanming Deng, Yangyi Huang, Xiaoyu Shi, Lewei Lu, Wenxiu Sun, Xiaogang Wang, Jifeng Dai, and Hongsheng Li. Fuseformer: Fusing fine-grained information in transformers for video inpainting. In Proceedings of the IEEE/CVF International Conference on Computer Vision, pages 14040–14049, 2021.
> >
> > [3] Zhen Li, Cheng-Ze Lu, Jianhua Qin, Chun-Le Guo, and Ming-Ming Cheng. Towards an end-to-end framework for flow-guided video inpainting. In Proceedings of the IEEE/CVF Conference on Computer Vision and Pattern Recognition, pages 17562–17571, 2022.430
> > [4] Gao, Chen, et al. "Flow-edge guided video completion." Computer Vision–ECCV 2020: 16th European Conference, Glasgow, UK, August 23–28, 2020, Proceedings, Part XII 16. Springer International Publishing, 2020.
> >
> > [5] Zou, Xueyan, et al. "Progressive temporal feature alignment network for video inpainting." Proceedings of the IEEE/CVF Conference on Computer Vision and Pattern Recognition. 2021.
> >
> > [6] Xu, Rui, et al. "Deep flow-guided video inpainting." Proceedings of the IEEE/CVF Conference on Computer Vision and Pattern Recognition. 2019.

---

> > ### Author Response · Authors · 2023-08-29
> > **Further response to reviewer BsWN**
> >
> > ### Q2 & Q3.More advanced video inpainting methods and comparisons
> >
> > Following you valuable feedback, since we have already compared with all popular and SOTA end-to-end video inpainting methods. So, we furtehr test more non-end-to-end (multi-stage) methods, they are the latest methods with code avaliable in ECCV'22. Please kindly check our preliminary comparison below.
> >
> > | Method | PSNR $\uparrow$ | SSIM $\uparrow$ | LPIPS $\downarrow$ | VFID $\downarrow$ | Runtime s/frame |
> > | ------ | ------ | ------ | ------ | ------ | ------ |
> > | FGT [1] | **31.5407** | 0.8967 | 0.0486 | 0.3368 | ~1.97 |
> > | ECFVI [2] | 20.8676 | 0.7692 | 0.0705 | 0.3019 | ~2.24 |
> > | **BSCVR-S (Ours)** | 28.8288 | ***0.9138*** | ***0.0399*** | **0.1704** | **0.172** |
> > | **BSCVR-P (Ours)** | ***29.0186*** | **0.9166** | **0.0391** | ***0.1730*** | ***0.178*** |
> >
> > These new quantitaive comparisons prove that non-end-to-end methods have significant computational complexity and requires much longer time consumption due to multi-stage framworks. For most of metrics, our methods can outperform them. The multi-stage framework is also hard for training and deployment [3]. Therefore we focus on end-to-end method and plan to update the full quantitative evaluation reults in our camera-ready version upon acceptance. We believe our research incorporating both end-to-end and non-end-to-end methods could provide more comprehensive benchmark for the related communities
> >
> > For qualitative comparison, we are making full visualization including these new methods and considering update those reults in our camera-ready version upon acceptance. Please kindly check our homepage for preliminary comparison.
> >
> > **Reference**
> >
> > [1] Kang, J., Oh, S. W., & Kim, S. J. (2022, October). Error compensation framework for flow-guided video inpainting. In European Conference on Computer Vision (pp. 375-390). Cham: Springer Nature Switzerland.
> >
> > [2] Zhang, K., Fu, J., & Liu, D. (2022, October). Flow-guided transformer for video inpainting. In European Conference on Computer Vision (pp. 74-90). Cham: Springer Nature Switzerland
> >
> > [3] Li, Zhen, et al. "Towards an end-to-end framework for flow-guided video inpainting." Proceedings of the IEEE/CVF conference on computer vision and pattern recognition. 2022.

---

> > > ### Author Response · Authors · 2023-08-30
> > > **Further response to reviewer BsWN**
> > >
> > > ### Q: More quanlitative comparisions are required to clearify the differences between the general video inpainting and the proposed dataset.
> > >
> > >
> > >
> > > 1. Besides our quantitative comparisons with recent advanced video inpainting methods in our last response, we further perform more qualitative comparisons to intuitively compare recovery quality of all methods. Please check our newly uploaded recovery videos in our github homepage. Please also note that it may need some time to load these videos for synchronous displays when accessing the homepage.
> > >
> > >
> > >
> > > 2. We can conclude that:
> > >
> > >     2.1 For general video inpainting methods pre-trained on general video inpainting dataset, they cannot achieve good performance in recovering real bitstream-corrupted videos. Please see STTN, FuseFormer, and E2FGVI-HQ in Table 2 of our paper, and FGT, ECFVI in our github homepage.
> > >
> > >
> > >
> > >     2.2 For general video inpainting methods trained on the proposed dataset, our dataset can help these methods improve their recovery performance in terms of all metrics. Please see STTN*, FuseFormer*, E2FGVI-HQ* in Table 2 of our paper.
> > >
> > >
> > >
> > >     2.3 In any case, our proposed method outperforms all general video inpainting methods no matter they are pre-trained on general video inpainting dataset or trained on our proposed dataset. Please see the Table 2 in our paper and our github homepage.
> > >
> > >
> > >
> > >     2.4 It demonstrates the value of our dataset and the advantage of our method in solving the problem of real bitstream-corrupted video recovery.
> > >
> > >
> > >
> > > Due to the time limit, we would carefully clarify the differences between the general video inpainting and the proposed dataset and provide more comprehensive results and detailed analysis in camera-ready version upon acceptance. Hope we have well solved all your concerns. We would appreciate your further feedback to make this research more solid and comprehensive for a higher score favoring acceptance.

---

### Official Review · Reviewer_rSJL · 2023-07-20
**Review of BSCV**

**Rating:** 7
**Confidence:** 4
**Correctness:** Yes.
**Clarity:** Yes.

**Strengths:**

- S1. The authors propose to randomly remove segments in bitstreams to simulate the effect of packet loss error and storage damage error on decoded videos. Such a simulation is more suitable for BVCR.

- S2. The size of the collected dataset is sufficient.

- S2. A powerful and simple baseline method is proposed.

**Additional Feedback:**

The authors should address my concerns on **simulation** and **resolutions**. Besides, I also have some questions:

- Q1. It is possible to get some real BVCR videos? What is the difference between real and simulated (including your simulated) videos?

- Q2. Large-scale data has shown great potential in both low-level and understanding tasks. Will generating more videos benefit BVCR task? Are 16k videos sufficient?

**Documentation:**

Yes.

**Ethics:**

No.

**Limitations:**

As mentioned above, the major two limitations of the submission are **simulation** and **resolutions** from my point of view. The authors should address my concerns.

**Opportunities For Improvement:**

- W1. Although the simulated method of this paper is better than typical works, the constructed dataset is still a simulated one. The authors should justify the difference between the simulated dataset and the real-world one.

- W2. Higher-resolution videos are a trend in the current era. The authors should also consider adding videos with higher resolutions (like 1080P, 2K) into the dataset.

**Relation To Prior Work:**

Yes.

**Summary And Contributions:**

The contributions of the submission are two-fold:

1. Collect a non-simulated dataset for bitstream-corrupted video recovery (BCVR).

2. Propose a simple yet strong baseline for BCVR.

---Update---

The authors have provided 1) results on high-resolution videos and H.265; 2) more baseline results; and 3) the necessity of the task. My concerns are addressed and I have no major reasons to reject this paper. Ergo, I still tend to accept this paper.

I have updated my scores -- from 6 -> 7. :)

---

> ### Author Response · Authors · 2023-08-22
> **Response to reviewer rSJL**
>
> Thank you for your feedback. Here we would like to discuss with you the issues you mentioned.
>
>
>
> ### Q1: Although the simulated method of this paper is better than typical works, the constructed dataset is still a simulated one. The authors should justify the difference between the simulated dataset and the real-world one.
>
> A1: We add more explanations to clarify the differences among simulated corruption, our corruption and real-world corruption. We also updated Fig.1 of the paper to justify the reality of our dataset and the similarity and difference between our dataset and the corrupted video in the real world.
>
> (1) The motivation of this work is the lack of video dataset decoded from corrupted bitstream, whereas bitstream-packet-loss-caused video corruption is a long-standing and common problem in video communications.
>
> (2) Traditionally, video errors in computer vision research are mostly simulated by manual-designed error patterns, e.g., blocks, slices, and masks. These corruption patterns are imposed on video frames in image domain. Our methods can better fit the bitstream packet loss by directly corrupting video bitstream by removing random bitstream fragments.
>
> (3) In the real world, packet loss typically leads to corrupted video with various corruption patterns. The lost packet is usually located randomly with hundreds and thousands of bytes. Thus, in bitstream domain, our model and method are matched with the packet loss in real world while making it controllable.
>
> (4) However, there exists some differences between our dataset and real-world one. For example, our corruption method skips the start code and headers in bitstreams, the corruption on these parts may cause severe errors, e.g., sequential frame missing, even decoding failures. Please refer to our added explanation in line 158 of the paper.
>
>
>
> ### Q2: Higher-resolution videos are a trend in the current era. The authors should also consider adding videos with higher resolutions (like 1080P, 2K) into the dataset.
>
> A2: (1) YouTube-VOS dataset contains a small number of higher resolution videos (e.g., 26bc786d4f, aa175e5ec7, fe6c244f63, 7eb2605d96, etc. in training set. 3d6a761295 in test set), and we have successfully encoded and manipulated those 1080P videos. With different branches, our dataset originally includes about 100 1080P video clips, 10,000 frames from YouTube-VOS. This proves the feasibility of our model and the resolution diversity of our dataset.
>
> (2) Following your suggestions, we additionally provide new subsets from YouTube-UGC and Videezy4K datasets. The YouTube-UGC subset contains over 250 videos under 1080P resolution, and Videezy4K subset provides 4K resolution. Please refer to the dataset link and check the “YouTube-UGC” and “Videezy4K” folders. And we update the manuscript by adding some discussion, please refer to line 174 of the paper as well as the table below.
>
> |Subset | DAVIS		| YouTube-VOS 	| YouTube-UGC	| Videezy4K 	|
> |  ----  |  ----  | ----  |  ----  | ----  |
> | Resolution | 480P 		| 720P        	| 1080P      	| 4K      	|
> |  Number of Videos per Branch | 150		| 4,000+			| 250+			| 12			|
> | Number of Branches | 7		| 7		| 1 (new branch gen. on going)			| 1			|
> |Function | test		| train/test		| (train)/test	| test		|
>
> ### Q3. It is possible to get some real BVCR videos? What is the difference between real and simulated (including your simulated) videos?
>
> A3: (1) We cannot find publicly available BCVR video dataset, but we found some BVCR examples and visualized them in Fig.1. We add more explanations to clarify the differences among simulated corruption, our corruption and real-world corruption and the reality of our dataset. Please refer to page 1 of the paper.
>
> (2) From the perspective of corruption way, our constructed BCVR videos have slight difference from the real-world one. For example, our corruption method skips the start code and headers in bitstreams, the corruption on these parts may cause severe errors, e.g., sequential frame missing, even decoding failures. In such cases, BCVR videos are not recovered by using vision-based methods. Please refer to line 158 of the paper.
>
> ### Q4. Large-scale data has shown great potential in both low-level and understanding tasks. Will generating more videos benefit BVCR task? Are 16k videos sufficient?
>
> A4: (1) Our dataset is currently sufficient for inpainting-based methods since 3,000+ 240p videos can meet their training requirements. Our dataset has updated with far more frames and the currently largest Vimeo90K, and it can still be sufficient for future research.
>
> (2) Our generated branches and newly added data source can enlarge the size of dataset, diversify the corruption types and degrees, and enrich the dataset context. Currently our dataset contains over 28K differently corrupted videos under different resolutions. With those more generated videos, BCVR task can also benefit from them.

---

> ### Comment · Reviewer_rSJL · 2023-08-26
> **Further concerns on this paper**
>
> Hi authors,
> Thank you for providing responses. I've read and checked your responses and the revised paper. I am glad to see some of my concerns are addressed. However, before I finalize my final recommendation, I think there are still some concerns to address.
>
> - C1. As pointed out by Reviewer BsWN, more related approaches should be re-implemented and evaluated. As a top-tier venue, a benchmark paper should be as comprehensive as possible. Ergo, the authors should provide more experimental results like typical video recovery methods.
>
> - C2. Reviewer rrQM also challenges the value of the paper. I think the author should also highlight the application value of the paper. For example, is studying the bitstream itself more valuable? Can we detect the corruption bitstream for retransmission?
>
> - C3. It is interesting to discuss the corruption in the start code and headers. Can the SOTA methods solve these corruptions? Should we solve this task in a computer vision manner? Can you further discuss this point?
>
> - C4. New results on higher resolution and H.265 are not included in the revised manuscript.

---

> > ### Author Response · Authors · 2023-08-28
> > **Response for further concerns to reviewer rSJL [1/2]**
> >
> > Thanks for your kind feedback and we are glad to discuss with you about your further concern.
> >
> > **A1:** Following your valuable suggestion, we perform more experiments with more methods.
> >
> > 1. Since we have already compared with all popular and SOTA end-to-end video inpainting methods. So, we further tested the latest non-end-to-end video inpainting methods including ECFVI [1] and FGT [2] from ECCV'22.
> >
> > 2. These non-end-to-end methods using traditional multi-stage framework require more computational resources and time consumption for processing data. Therefore, they have low computational efficiency and they can only process fixed and small resolutions (240P) [1, 2].
> >
> > 3. An preliminary evaluation result is shown in the Table, we test the latest method with their pre-trained model on part of our test set due to time-limitation.
> >
> > | Method | PSNR $\uparrow$ | SSIM $\uparrow$ | LPIPS$\downarrow$ | VFID$\downarrow$ | Runtime |
> > | ------ | ------ | ------ | ------ | ------ | ------ |
> > | FGT[2] | **31.5407** | 0.8967 | 0.0486 | 0.3368 | ~1.97 |
> > | ECFVI[1] | 20.8676 | 0.7692 | 0.0705 | 0.3019 | ~2.24 |
> > | **BSCVR-S (Ours)** | 28.8288 | ***0.9138*** | ***0.0399*** | **0.1704** | **0.172** |
> > | **BSCVR-P (Ours)** | ***29.0186*** | **0.9166** | **0.0391** | ***0.1730*** | ***0.178*** |
> >
> > 4. The latest methods still cannot outperform our methods even they require significantly more time for data processing. We would like to further highlight the limitation of those non-end-to-end multi-stage frameworks by updating the full results and explanation in camera-ready version upon acceptance.
> >
> > **A2:**
> >
> > 1. Our understanding is that Reviewer rrQM just has some concerns about the matching of our paper to the scope of NeurIPS, instead of challenging the value of the paper.
> >
> > 2. The application value of the paper can be regarded as the **last resort** to improve the video quality in video communications.
> >
> >     2.1 To deal with video bitstream corruption (packet loss), some bitstream processing and image processing techniques are used in practice to help enhance the end users’ visual fidelity.
> >
> >     2.2 Lost data can be retransmitted in some bidirectional channels with feedback, but this approach is not always feasible due to the inherent delay or lacking feedback in unidirectional channels [3].
> >
> >     2.3 Channel coding is another technique by which the data are protected using Forward Error Correction (FEC) codes. Source coding with/without channel coding is known as Error Resilient Coding (ERC). It exploits source coding tools to make the compressed video more resilient against channel errors/losses [4, 5].
> >
> >     2.4 Although FEC and ERC coded videos can resist errors and losses to some extent, there are scenarios where the received video is too erroneous and still needs further processing [3].
> >
> >     2.5 Due to the predictive nature of the video codecs, the concealment distortion is easily propagated to the future frames which sometime make the video totally unusable [3]. In other word, even though bitstream can get mostly recovered, but the decoded video errors can still be severe.
> >
> >     2.6 In the practical applications, video is captured->compressed (source coding)->transmitted (channel coding)->display (vision domain). Therefore, concealing errors, e.g., by video inpainting, and error concealment can be regarded as the **last resort** to improve the video quality in case channel coding and source coding are not efficient.
> >
> >     2.7 However, existing error/mask patterns, e.g., fixed blocks, slices, or manual-designed masks are remarkably different from the practical damaged video bitstream, illustrated as the redrawn Fig. 1. of the paper.
> >
> >     2.8 To rethink the gap between existing restoration/inpainting methods and practical video corruption characteristics, we constructed the first bitstream corrupted video (BSCV) benchmark dataset and developed method to improve quality from practical video corruption.
> >
> > 3. To summarize, video is captured, compressed, transmitted, and finally for display purposes. Thus,
> >
> >     3.1 Video recovery in vision domain no need channels of bidirectional communications, and this approach is more intuitive for users of video.
> >
> >     3.2 Our constructed benchmark dataset can reflect the practical video characteristics in video communications. Our method can help enhance end users’ visual fidelity by recovering practical corrupted videos.
> >
> >     3.3 Therefore, this paper has great application value and prospect for most of the popular video scenarios, e.g., internet videos (YouTube, NetFlix), living stream (TikTok), and video conference (Zoom), and other video communications, where bitstream is possible corrupted.
> >
> >     3.4 In our final camera-ready paper, we will highlight the application value of this research.

---

> > ### Author Response · Authors · 2023-08-28
> > **Response for further concerns to reviewer rSJL [2/2]**
> >
> > **A3:**
> >
> >
> >
> > 1. It is indeed an interesting topic when the corruption in the start code and headers. However, since the ratios of them to a bitstream are too slight (e.g., 10-5 scale). It means negligible corruption probability. So far, there are no public related literature to discuss this.
> >
> >
> >
> > 2. We have made a preliminary investigation to study different parts’ corruption will result in what kind of consequences.
> >
> >
> >
> >     2.1 Start code prefix (3 bytes, e.g., 0x 000001) is like delimiter between NALUs (network abstraction layer units), NAL Header (1 bytes, e.g., 0x 65) is like data type flag for a NALU.
> >
> >
> >
> >     2.2 Corrupting start code and header of NALU (e.g., i-th), may cause the decoded video sequences lose/skip this frame, and the following frames may have regional corruption or repeat the former correctly decoded frame (i-th – 1 ).
> >
> >
> >
> >     2.3 Corrupting start code of SPS and PPS (special NALU that saves global parameters), may cause decode failure, no frame of the video can be decoded.
> >
> >
> >
> >     2.4 To summarize, the corruption on these parts may cause severe errors, e.g., continuous missing of multiple frames, even decoding failures, which is not a video inpainting problem. If frame missing is not severe, it may consider to use computer vision techniques, e.g., frame interpolation. If decoding failures, it may not be well solved in a computer vision manner.
> >
> >
> >
> > 3. We are aware that such investigation needs more solid assumption, rich analysis, extensive validation, and careful induction. In our final camera-ready paper, we can present our investigation. In the future, we may consider conducting a more systematical and comprehensive investigation to benefit related research.
> >
> >
> >
> > **A4:**
> >
> >
> >
> > 1. We conduct this heavy task on the newly-generated 1080P subset encoded by H.265 protocols. We would like to share this difficult process and our initial evaluation results.
> >
> >
> >
> > 2. It should be mentioned that processing videos in 1080P resolution is a very challenging problem, because learning-based video inpainting is currently a computationally heavy task [4]. Only our method and the SOTA E2FGVI-HQ method can handle video more than 720P. Thus, our validation experiments are performed on this two methods.
> >
> >
> >
> > 3. As the SOTA E2FGVI-HQ method states [6], processing 720P video from YouTube-VOS dataset requires GPU with 48G VRAM. For most devices of common researchers, this can be regarded as a very strict hardware requirement.
> >
> >
> >
> > 4. We managed to using 24G VRAM to process 1080P video by code optimization and frame rate downsampling. Please see the Table.
> >
> >
> >
> > 5. The 1080P + H.265 generates quite low quality videos due to more severe error propagation caused by the more complex prediction dependencies of H.265. Although all methods perform not as good as in low resolution case, but our method could still outperform the SOTA E2FGVI-HQ method in terms of most metrics.
> >
> >
> >
> > | Method | PSNR $\uparrow$ | SSIM $\uparrow$ | LPIPS $\downarrow$ |
> > | ------ | ------ | ------ | ------ |
> > | Input | ~9.9011 | ~0.1629 | ~0.4629 |
> > | E2FGVI-HQ I | 17.4366 | 0.6396 | 0.2129 |
> > | **BSCVR-S (Ours)** | 16.9523 | 0.6438 | 0.2100 |
> >
> > **Reference**
> >
> > [1] Kang, J., Oh, S. W., & Kim, S. J. (2022, October). Error compensation framework for flow-guided video inpainting. In European Conference on Computer Vision (pp. 375-390). Cham: Springer Nature Switzerland.
> >
> >
> >
> > [2] Zhang, K., Fu, J., & Liu, D. (2022, October). Flow-guided transformer for video inpainting. In European Conference on Computer Vision (pp. 74-90). Cham: Springer Nature Switzerland.
> >
> >
> >
> > [3] Kazemi, M., Ghanbari, M., & Shirmohammadi, S. (2021). A review of temporal video error concealment techniques and their suitability for HEVC and VVC. Multimedia Tools and Applications, 80, 12685-12730.
> >
> >
> >
> > [4] Fleury M, Altaf M, Moiron S, Qadri N, Ghanbari M (2012) Source coding methods for robust wireless video streaming. Multimedia Networking and Coding:175–207
> >
> >
> >
> > [5] Wang Y, Wenger S, Wen J, Katsaggelos AK (2000) Review of error resilient coding techniques for realtime video communications. IEEE Signal Processing Magazine 1(4):61–82
> >
> >
> >
> > [6] Li, Zhen, et al. "Towards an end-to-end framework for flow-guided video inpainting." Proceedings of the IEEE/CVF conference on computer vision and pattern recognition. 2022.

---

> > > ### Comment · Reviewer_rSJL · 2023-08-29
> > >
> > > Hi authors,
> > >
> > > I appreciate your new justifications. My concerns are well addressed. It seems that it is valuable to study this task since both current computer vision models and FEC/ERC techniques can not well address this challenging task. A minor weakness is that the performance of the proposed baseline is not powerful compared with FGT and ECFVI. The authors should further explain the limitations of these methods.

---

> > > > ### Author Response · Authors · 2023-08-29
> > > > **Further response to reviewer rSJL**
> > > >
> > > > ### Q: A minor weakness is that the performance of the proposed baseline is not powerful compared with FGT and ECFVI. The authors should further explain the limitations of these methods.
> > > >
> > > >
> > > >
> > > > Thank you for increasing your score from 6 to 7. Your valuable suggestions make this paper more solid and comprehensive.
> > > >
> > > >
> > > >
> > > > 1. For the quantitative evaluation, even though our method didn’t obtain the highest PSNR, but it achieves the best SSIM, LPIPS (perceptual metric), and VFID (perceptual metric) than newly evaluated non-end-to-end methods, FGT and ECFVI.
> > > >
> > > > 2. We further perform qualitative evaluation for visual quality comparison. Some visualizations are illustrated in our dataset homepage. It demonstrates that our method could achieve the best visual quality.
> > > >
> > > > 3. These non-end-to-end methods has limitations including:
> > > >
> > > >     3.1 They are computationally heavy and time-consuming, which costs more than 10 times than our method.
> > > >
> > > >     3.2 They failed to generate good recovery with high visual fidelity, which may not be able to satisfy the demand of users for pleasant visual experience.
> > > >
> > > >     3.3 The multi-stage training strategy adopted in non-end-to-end methods greatly increase the difficulties in terms of training, transplantation, and deployment for broader applications [1].
> > > >
> > > >      3.4 For the multi-stage structure, error/inaccurate information in early stages may cause severe propagations to the late stages [1], resulting in unsatisfactory recovery quality.
> > > >
> > > > We would like to summarize these limitations of non-end-to-end methods, and update the evaluation results in camera-ready version upon acceptance. It can enhance the rationality in our baseline selection.
> > > >
> > > > **Reference**
> > > >
> > > > [1] Li, Zhen, et al. "Towards an end-to-end framework for flow-guided video inpainting." Proceedings of the IEEE/CVF conference on computer vision and pattern recognition. 2022.

---

> > > > > ### Comment · Reviewer_rSJL · 2023-08-29
> > > > >
> > > > > From my point of view, PSNR lacks readability. A better PSNR does not mean better visualization since the model will oversmooth the video to purchase a high PSNR. Therefore, I treat the PSNR performance as a minor problem. By the way, I have checked your visual cases. Can you additionally provide some video/gif performance comparison in your github so that I can better check the visualization result? I think a video form presentation can better highlight the model advantages.

---

> > > > > > ### Author Response · Authors · 2023-08-30
> > > > > > **Further response to reviewer rSJL**
> > > > > >
> > > > > > ### Q: By the way, I have checked your visual cases. Can you additionally provide some video/gif performance comparison in your github so that I can better check the visualization result? I think a video form presentation can better highlight the model advantages.
> > > > > >
> > > > > >
> > > > > >
> > > > > > 1. Thanks for the further comments. Following your suggestion, we additionally provide some video/gif performance comparison in github homepage. Please also note that it may need some time to load these videos for synchronous displays when accessing the homepage.
> > > > > >
> > > > > >
> > > > > >
> > > > > > 2. For your convenience in the video checking, please notice the details include:
> > > > > >
> > > > > >     Scene 1. The mascot being flipped, the fast-moving players and the ground of the stadium/court.
> > > > > >
> > > > > >     Scene 2. The stability of the structure of the suit, and the facial details of the man.
> > > > > >
> > > > > >     Scene 3. The body of the tennis player.
> > > > > >
> > > > > >
> > > > > >
> > > > > > 3. These video form presentations well highlight the advantages of our model in recovering long-term video sequences and large area error patterns caused by bitstream corruptions.
> > > > > >
> > > > > >
> > > > > >
> > > > > > We would thank you again for your valuable comments and always quick feedback. If you have any further concerns that are helpful for us in obtaining a higher score, we would be very glad to address them for the final acceptance of this paper.

---

> > > > > > > ### Comment · Reviewer_rSJL · 2023-08-30
> > > > > > >
> > > > > > > I have checked the gifs. I am satisfied with the visual results. The proposed baseline recovers the video better. I have no further concerns on this paper. My final rating is 'Accept'.

---

### Official Review · Reviewer_f8pj · 2023-07-21
**Timely dataset for bitstream-corrupted video recovery**

**Rating:** 8
**Confidence:** 3
**Correctness:** The claims made in the paper are corr…
**Clarity:** Yes, the paper is well written.

**Strengths:**

This is an important question. Compared to the previous method of simulating errors with manually designed masks, the use of bitstream to construct the dataset by the authors is indeed more representative of real-world application scenarios. The experiments are quite comprehensive. The paper is well-written and easy to understand.

**Additional Feedback:**

Is your method applicable to H265? If you want to support other updated protocols, will it require a significant effort?

**Documentation:**

Yes

**Limitations:**

I don't think the paper has any obvious limitations.

**Opportunities For Improvement:**

here are some minor issues, such as missing page numbers, but these small details don't affect my enjoyment of reading the paper.

**Relation To Prior Work:**

Yes

**Summary And Contributions:**

The authors propose a bitstream-corrupted video recovery dataset and benchmark. Instead of simulating the missing content by manual-designed error masks, the authors introduce the bitstream-corrupted video benchmark and dataset. The BSCV is a collection of 1) a proposed three-parameter corruption model for video bitstream, 2) a large-scale dataset containing rich error patterns, multiple corruption levels, and flexible dataset branches, and 3) a plug-and-play module in video recovery framework that serves as a benchmark. We evaluate state-of-the-art video inpainting methods on the BSCV dataset, demonstrating existing approaches' limitations and our framework's advantages in solving the bitstream-corrupted video recovery problem.

---

> ### Author Response · Authors · 2023-08-22
> **Response to reviewer f8pj**
>
> Thank you for your comment. We would like to respond to your concern here.
>
> ### Q: Is your method applicable to H265? If you want to support other updated protocols, will it require a significant effort?
>
> A: Our method is also applicable to H.265 and does not require significant effort since H.265 is developed based on H.264 and they have similar bitstream structure. Specifically, to apply our method to other video encoding protocols, it will share similar procedure as we introduced in the paper.
>
> (1) We first need to use video codec to encode the target video files to the bitstream coded by the specific protocol. It is possible that our provided H.264 bitstream can be recoded to H.265 using the existing video codec.
>
> (2) Then, we need to read the bitstream and manipulate it by programs. The program should firstly read the bitstream, then locate the content bitstream by searching specific start code to set the possible corruption location.
>
> (3) Next, with configured parameters, several fragments could be deleted and the rest bitstream will be concatenated to form a corrupted bitstream. Then using the video codec again to decode the corrupted bitstream to generate the new dataset.

---

> > ### Author Response · Authors · 2023-08-26
> > **Response to reviewer f8pj about new H265 subset**
> >
> > Following your feedback, we further test our method with H.265 protocol on 4K videos, and we generate another 4KH265 subset. It demonstrates that our method is also applicable to updated protocols which requires no significant efforts.

---

### Official Review · Reviewer_gEor · 2023-07-21
**a good realistic corruption construction but still needs improvment.**

**Rating:** 5
**Confidence:** 4
**Correctness:** yes.
**Clarity:** yes.

**Strengths:**

This paper is well-written and the dataset contains good diversity of corruption patterns and dataset branches. The authors also propose a module that could be used in the video recovery framework. The dataset has realistic patterns rather than some simulated corruption patterns, which is convincing.

**Additional Feedback:**

No.

**Documentation:**

yes.

**Ethics:**

Don't know.

**Limitations:**

please see the Opportunities For Improvement

**Opportunities For Improvement:**

1. Although the authors list many hard and realistic corruption patterns in Figure1, but the demonstrated results have similar corruption patterns.
2. The source of the dataset is only from two datasets, I expect it from more kinds of dataset.
3. The paper should also state or use some metrics to show how their corruption way is close to the real corruption happening in the real world.
4. The proportion of different types of corruption is still not varied shown in Figure 3 and why the experimental setting is set the parameter to (1/16, 0.4, 4096). The flexibility part in dataset construction is not convincing to me because there are only 4 settings with slight changes in parameters. The final experimental results don't exceed E2FGVI-HQ a lot.
5. The paper didn't include a checklist.

**Relation To Prior Work:**

yes.

**Summary And Contributions:**

This paper constructs the first large-scale dataset used for bitstream-corrupted video recovery. The dataset contains different types of realistic corruption patterns. Then the authors propose a plug-and-play module to embed in video inpainting frameworks to better capture the feature of corrupted videos. Finally, the authors do a comprehensive evaluation to reveal existing video inpainting methods based on their new dataset.

---

> ### Author Response · Authors · 2023-08-22
> **Response to reviewer gEor [1/3]**
>
> Thank you for your comments on our work. We have responded to all your concerns below.
>
>
>
> ### Q1: Although the authors list many hard and realistic corruption patterns in Figure1, the demonstrated results have similar corruption patterns.
>
> A1: For the demonstrated results, we redraw Fig.5. We additionally visualized the corruption patterns at the leftmost column. It involves all the corruption patterns and their complex combinations. The diverse patterns are harder and more realistic than those in Fig.1. Please refer to our redrawn Fig.5 and explanation in line 295 of the paper.
>
>
>
> ### Q2: The source of the dataset is only from two datasets, I expect it from more kinds of dataset.
>
> A2: (1) YouTube-VOS and DAVIS are popular large-scale benchmark datasets for video inpainting tasks, the recent methods, e.g., STTN[5], FuseFormer[6], E2FGVI[7] and more prior works, e.g., FGVC[8], TSAM[9], DFVI[10] are commonly trained on YouTube-VOS, and tested on DAVIS. Therefore, we follow these general practices for a fair comparison.
>
> (2) Following your suggestion, we also have expanded dataset sources and enlarged dataset scales. In detail, we additionally select YouTube-UGC [11] and Videezy4K [12] datasets as new data sources. The video on YouTube-UGC has larger frame numbers and 1080P resolution. Videezy4K contains fewer frame numbers and amuch larger resolution (4K) than YouTube-VOS and YouuTube-UGC. The setting of corruption parameter combination, their statistics are shown below, more branches is generating and uploading
>
> | (2/16,0.4,4096)	|   YouTube-VOS 	| YouTube-UGC | Videezy4K	|
> | ------ | ------ | ------ | ------ |
> | Resolution     |  720P      		| 1080P      | 4K      	|
> | Uncorrupted    | ～40%       	| 41.5%      | 30%     	|
> | Minor          | ～25%       	| 7.5%       | 5.4%    	|
> | Moderate       | ～20%       	|  11.7%      | 19.2%   	|
> | Severe         | ～15%      	| 39.3%      | 45.4%   	|
>
> Therefore, the more kinds of resolutions, frame numbers, and data source make our constructed dataset more diverse and comprehensive.
>
>
>
> ### Q3: The paper should also state or use some metrics to show how their corruption way is close to the real corruption happening in the real world.
>
> A3: Thanks for the constructive suggestion, we redrawn the illustration of corruption case comparison in Fig. 1 with collected real examples and we also add more explanation about our corruption model.
>
> (1) **Authenticity of corruption pattern in the image domain:** We redrawn Fig,1 to better clarify the difference and similarities between simulated corruption, real corruption, and our generated corruption. It shows that our corruption patterns are closer to reality in the image domain.
>
> (2) **Rationality of the corruption model in the bitstream domain:** In the application scenario of living stream, video conference, and other video communications, packet loss issue is a typical corruption which may cause by the low-bandwidth and unstable channel. As [1] introduced, packet loss rate, lost packet size, and location are three main factors directly affecting the quality of videos. Then referring to the common network settings with packet size 800-1500 bytes [4], we can configure these parameters, unlike prior research [2][3] using single parameter simulation. Our proposed corruption model can well fit the packet loss by removing random segments in bitstream.

---

> > ### Author Response · Authors · 2023-08-22
> > **Response to reviewer gEor [2/3]**
> >
> > ### Q4: The proportion of different types of corruption is still not varied shown in Figure 3 and why the experimental setting is set the parameter to (1/16, 0.4, 4096). The flexibility part in dataset construction is not convincing to me because there are only 4 settings with slight changes in parameters. The final experimental results don't exceed E2FGVI-HQ a lot.
> >
> > A4: We further extended the explanation of Fig. 3 and Fig. 6. In Fig. 6, we provide more parameter combinations to demonstrate the effectiveness of our model.
> >
> > (1) **Unvaried proportion in Fig.3.** Our dataset contains different branches (by varied parameter settings) with different subsets (for training and testing), involving different corruption ratios.
> >
> > - To make the corruption ratio controllable, the model is required to generate consistent results among different subsets.
> >
> > - Fig. 3(a) is therefore used to illustrate that in one branch, our corruption model can generate consistent damage ratios among different subsets, which shows the balance of our dataset and effectiveness of our model.
> >
> > (2) **The reason for (1/16, 0.4, 4096).** By configuring different parameter combinations of our corruption model, we can generate multiple dataset branches.
> >
> > - The quality of the corrupted video of all branches is measured by all the metrics in Tab.3 (The rows of “Input”). It shows that the (1/16, 0.4, 4096) branch has the best PSNR, SSIM, LPIPS, and second-best VFID. Therefore, we select it as the base for training and the rest branches for testing.
> >
> > - This experimental setting is to evaluate the robustness of our method when the corruption is getting severer. Also, we use it to verify the effect of parameter adjustment on recovery performance.
> >
> > (3) **More changes in parameters.** Following your suggestion, we additionally constructed 3 new branches by setting more significant parameter changes. Totally, 7 branches with greatly varied proportions are shown in Fig.6 and Tab.3. These branches greatly improved the diversity of corruption severity, producing more tough scenarios for recovery. Our method is still more effective than the comparison method in those tough branches shown in Tab.3.
> >
> > | Param.          | Method 	| PSNR $\uparrow$ 		| SSIM $\uparrow$    | LPIPS $\downarrow$   | VFID $\downarrow$    |
> > | ------ | ------ | ------ | ------ | ------ | ------ |
> > | ...             | ...    	| ...    	| ...    | ...    | ...    |
> > | (1/16,0.4,8192) | Input 	| 18.66665	| 0.8170 | 0.1450 | 0.1849 |
> > |                 | E2FGVI-HQ	| 26.0722 	| 0.8371 | 0.0486 | 0.0455 |
> > |                 | BSCVR-S 	| 26.2708 	| 0.8518 | 0.0423 | 0.0417 |
> > |                 | BSCVR-P 	| 26.1231 	| 0.8487 | 0.0430 | 0.0418 |
> > | (1/16,0.8,4096) | Input 	| 19.0062 	| 0.8419 | 0.1389 | 0.1874 |
> > |                 | E2FGVI-HQ	| 32.8311 	| 0.9506 | 0.0162 | 0.0264 |
> > |                 | BSCVR-S 	| 32.7959 	| 0.9514 | 0.0147 | 0.0247 |
> > |                 | BSCVR-P 	| 32.7204 	| 0.9509 | 0.0149 | 0.0252 |
> > | (4/16,0.4,4096) | Input  	| 17.8542 	| 0.7587 | 0.1610 | 0.1973 |
> > |                 | E2FGVI-HQ	| 22.7912 	| 0.7480 | 0.0738 | 0.1192 |
> > |                 | BSCVR-S 	| 22.7094 	| 0.7570 | 0.0679 | 0.1079 |
> > |                 | BSCVR-P 	| 22.5480 	| 0.7527 | 0.0686 | 0.1079 |
> >
> > (4) **Experimental results.** Regarding performance improvement, our contribution is not only the module but also the high-quality data.
> >
> > - Recovery of real corrupted video lacks research because the unrealistic corruption assumption limits the performance of video inpainting. In our experiments, the realistic corruption provided by our dataset helped the existing SOTA video inpainting method gain better performance with only half training iterations.
> >
> > - Therefore, our contribution should not be interpreted by comparing the last three columns. Instead, compared with the pre-trained E2FGVI-HQ, our high-quality data offers noticeable performance improvement with higher efficiency, then our module makes a further improvement, totally over 1.5dB in higher resolution videos.

---

> > > ### Author Response · Authors · 2023-08-22
> > > **Response to reviewer gEor [3/3]**
> > >
> > > ### Q5: The paper didn’t include a checklist.
> > >
> > > A5: Since NeurIPS 2023’s guideline and the template files indicated that “Note that this year the checklist will be entered in OpenReview, so you should not include it in the submission PDF. “, we submitted the form on OpenReview rather than attaching it in our paper.
> > >
> > >
> > >
> > > **Reference**
> > >
> > > [1] R. Aravind, M. R. Civanlar and A. R. Reibman, "Packet loss resilience of MPEG-2 scalable video coding algorithms," in IEEE Transactions on Circuits and Systems for Video Technology, vol. 6, no. 5, pp. 426-435, Oct. 1996, doi: 10.1109/76.538925.
> > >
> > > [2] Cheng-Han Lin, Chih-Heng Ke, Ce-Kuen Shieh and N. K. Chilamkurti, "The Packet Loss Effect on MPEG Video Transmission in Wireless Networks," 20th International Conference on Advanced Information Networking and Applications - Volume 1 (AINA'06), Vienna, Austria, 2006, pp. 565-572, doi: 10.1109/AINA.2006.325.
> > >
> > > [3] Niedermeier, Florian, Michael Niedermeier, and Harald Kosch. "Quality assessment of the MPEG-4 scalable video CODEC." Image Analysis and Processing–ICIAP 2009: 15th International Conference Vietri sul Mare, Italy, September 8-11, 2009 Proceedings 15. Springer Berlin Heidelberg, 2009.
> > >
> > > [4] Kim, Kwang Soon, et al. "Ultrareliable and low-latency communication techniques for tactile internet services." Proceedings of the IEEE 107.2 (2018): 376-393.
> > >
> > > [5] Yanhong Zeng, Jianlong Fu, and Hongyang Chao. Learning joint spatial-temporal transformations for video inpainting. In Computer Vision–ECCV 2020: 16th European Conference, Glasgow, UK, August 23–28, 2020, Proceedings, Part XVI 16, pages 528–543. Springer, 2020
> > >
> > > [6] Rui Liu, Hanming Deng, Yangyi Huang, Xiaoyu Shi, Lewei Lu, Wenxiu Sun, Xiaogang Wang, Jifeng Dai, and Hongsheng Li. Fuseformer: Fusing fine-grained information in transformers for video inpainting. In Proceedings of the IEEE/CVF International Conference on Computer Vision, pages 14040–14049, 2021.
> > >
> > > [7] Zhen Li, Cheng-Ze Lu, Jianhua Qin, Chun-Le Guo, and Ming-Ming Cheng. Towards an end-to-end framework for flow-guided video inpainting. In Proceedings of the IEEE/CVF Conference on Computer Vision and Pattern Recognition, pages 17562–17571, 2022.430
> > > [8] Gao, Chen, et al. "Flow-edge guided video completion." Computer Vision–ECCV 2020: 16th European Conference, Glasgow, UK, August 23–28, 2020, Proceedings, Part XII 16. Springer International Publishing, 2020.
> > >
> > > [9] Zou, Xueyan, et al. "Progressive temporal feature alignment network for video inpainting." Proceedings of the IEEE/CVF Conference on Computer Vision and Pattern Recognition. 2021.
> > >
> > > [10] Xu, Rui, et al. "Deep flow-guided video inpainting." Proceedings of the IEEE/CVF Conference on Computer Vision and Pattern Recognition. 2019.
> > >
> > > [11] Wang, Yilin, Sasi Inguva, and Balu Adsumilli. "YouTube UGC dataset for video compression research." 2019 IEEE 21st International Workshop on Multimedia Signal Processing (MMSP). IEEE, 2019.
> > >
> > > [12] Guo, Shi, et al. "A differentiable two-stage alignment scheme for burst image reconstruction with large shift." Proceedings of the IEEE/CVF Conference on Computer Vision and Pattern Recognition. 2022

---

> > > > ### Author Response · Authors · 2023-08-30
> > > > **Further response to reviewer gEor**
> > > >
> > > > ### Q4: The final experimental results don't exceed E2FGVI-HQ a lot.
> > > >
> > > >
> > > >
> > > > 1. To better visualize experimental results for intuitive comparisons, we have uploaded final recovered videos in our github homepage to directly compare the recovery quality of different methods. Please check the section Method in the github homepage. Please also note that it may need some time to load these videos for synchronous displays when accessing the homepage.
> > > >
> > > >
> > > >
> > > > 2. For your convenience in the video checking, please notice the details include:
> > > >
> > > >     Scene 1. The lower limbs of the running lady, the table on the right.
> > > >
> > > >     Scene 2. The water spray and diver's swimsuit in the lower half of the video.
> > > >
> > > >     Scene 3. The vehicle body and wheels are in the center of the video.
> > > >
> > > >     Scene 4. The body of the parked white and yellow car.
> > > >
> > > >     Scene 5. The grassland and the horse.
> > > >
> > > >     Scene 6. The road structure, the faces of two mans, the texture of the suite.
> > > >
> > > >
> > > >
> > > > 3. These comparisons provide a more intuitive presentation to show that
> > > >
> > > >     3.1. Our method has noticeable advantage over all comparison methods in visual quality. More structure/contour and detail/texture information can be recovered by our method. Meanwhile, our method can make sure the consistency/stability of recovery quality among long sequences of corrupted video.
> > > >
> > > >     3.2. Furthermore, training comparison methods using our dataset can effectively improve their performance in recovering real bitstream-corrupted videos, demonstrating the value of our dataset.
> > > >
> > > >     3.3 Our method still outperform all comparison methods even though they are also trained on our dataset, demonstrating the advantage of our proposed method.
> > > >
> > > >
> > > >
> > > > Thanks again for your valuable comments, which has significantly helped us in improving our manuscripts and dataset. We would appreciate your further feedback to make this research more solid and comprehensive.
> > > >
> > > >
> > > >
> > > > Hope we have well solved all your concerns. We would appreciate you could consider upgrading score to forward this research acceptance, so as to timely facilitate more research work in our research communities.

---

### Author Response · Authors · 2023-08-22
**Author response to ALL**

Dear reviewers, ACs and PCs,

We sincerely thank the reviewers for your time and effort in reviewing our paper and providing constructive comments that guided us in improving the quality of our manuscript. We are encouraged to receive the reviewers’ appreciation of our work from their contribution and strength summary in the following aspects:

- Proposing an important real world video recovery problem which lacks research. [Reviewer f8pj, BsWN, rrQM]

- The scale of the dataset is large and sufficient. [Reviewer: gEor, f8pj, rSJL BsWN, rrQM]

- Dataset contains more realistic and convincing corruption than previous task setting for video recovery [Reviewer gEor, f8pj, rSJL, BsWN]



- Data has good diversity [Reviewer gEor, f8pj]



- Writing is easy to understand [Reviewer gEor, f8pj, rrQM ]



- Useful and simple baseline method [Reviewer gEor, rSJL]



- East to access [Reviewer rrQM]



- Comprehensive experiments [Reviewer f8pj]



We have addressed all the concerns raised by reviewers and updated our manuscript. We would like to highlight the additional results suggested by the reviewers to demonstrate the high quality and expandability of our dataset。



- We generate additional branches on the current YouTube-VOS&DAVIS subset with more tough parameter combinations which produce more diverse data.



- We generate new subsets with more resolution including 1080P and 4K videos from YouTube-UGC and Videezy4K datasets to demonstrate the effectiveness of our dataset construction method. And the current dataset size is larger.



|Subset | DAVIS		| YouTube-VOS 	| YouTube-UGC	| Videezy4K 	|
|  ----  |  ----  | ----  |  ----  | ----  |
| Resolution | 480P 		| 720P        	| 1080P      	| 4K      	|
|  Number of Videos per Branch | 150		| 4,000+			| 250+			| 12			|
| Number of Branches | 7		| 7		| 1 (new branch gen. on going)			| 1			|
|Function | test		| train/test		| (train)/test	| test		|



- We conduct more qualitative and quantitative performance evaluations to better visualize the difference between bitstream-corrupted video recovery and traditional simulated video recovery problem.



Further expansion of the dataset, as well as uploading, is in progress, and the revised version of the manuscript represents our current results, which we will continue to update. We are willing to have further discussion and welcome any additional comments from reviewers.



Yours sincerely,



Authors

---

### Author Response · Authors · 2023-08-26
**Paper Revision**

We sincerely thank all reviewers and chairs again for constructive comments and feedback. Please find our updated Supplemenertary Materials ZIP file, we uploaded our revised version manuscript containing highlighted revisions, named "517_bitstream_corrupted_video_reco_RevisedManuscript_highlight.pdf". Our initial version manuscript named "517_bitstream_corrupted_video_reco_OriginalManuscript.pdf" is also included in the ZIP file for comparison.

For your convinience, all our revisions are listed in the following Table:

| Revision | Page & Line | For Reviewer |
| -------- | ------- | ------- |
| Expand dataset and update the statistics. | P1L7, P3L55&71 | gEor Q2/4, rSJL Q2/4 |
| Redraw Fig.1 to clarify differences. | P2 Fig.1 | gEor Q1/3, rSJL Q1/3 |
| Clarify the dataset construction method. | P5 L158 | gEor Q3, rSJL Q1/3 |
| Add dataset sources and more resolutions. | P5 L174 | gEor Q2, rSJL Q2 |
| Redraw Fig.4 with updated caption. | P7 Fig.4 | BsWN Q1/2 |
| New experiments for comparing pre-trained E2FGVI-HQ and our method. | P8 Tab.2 | gEor Q4 |
| Redraw Fig.5. to visualize corruption patterns and more qualitative evaluations. | P9 Fig.5 | gEor Q1/4, BsWN Q2/3  |
| Add according explanations for the redrawn Fig.5 and evaluations. | P9 L295 | gEor Q1/4, BsWN Q2/3 |
| Redraw Fig.6 for new branches with tougher parameter combinations. | P10 Fig.6 | gEor Q2/4, rSJL Q4 |
| Add according explanations for new dataset branches | P10 L312 | gEor Q2/4, rSJL Q4 |
| Expand Tab.3 to include more branches and evaluations. | P10 Tab.3 | gEor Q4, rSJL Q4 |
| Construct new dataset subset using 4K resolution videos and H.265 protocol. | Dataset url | f8pj, rSJL Q2 |

The paper could be further revised with the statistics and explanation updates because our large scale dataset (2TB) is continuously being expanded in terms of dataset sources, video resolutions, and codec protocols.

---

### Decision · Program_Chairs · 2023-09-22

**Decision:**

Accept (Poster)

**Comment:**

This paper introduces the Bitstream-Corrupted Video (BSCV) benchmark, a novel dataset comprising over 28,000 video clips, designed for real-world bitstream-corrupted video recovery. It presents a three-parameter corruption model, a large-scale dataset with diverse error patterns and corruption levels, and a plug-and-play recovery module for benchmarking. The study evaluates state-of-the-art video inpainting methods using the BSCV dataset, highlighting the limitations of existing approaches and the advantages of their framework in addressing bitstream-corrupted video recovery. Video recovery is an important topic. Despite some reviewers raising questions, such as the need for more extensive method comparisons, it does not diminish the contributions of this paper. The authors provided constructive and reasonable responses during the rebuttal phase, addressing the reviewers' concerns to some extent. Taking all factors into consideration, I believe this paper meets the acceptance criteria and recommend accepting it.